# Unlocking the Potential of Global Human Expertise

**Elliot Meyerson**[1]  **Olivier Francon**[1]  **Darren Sargent**[1]  **Babak Hodjat**[1]  **Risto Miikkulainen**[1,2]

[1]Cognizant AI Labs   [2]The University of Texas at Austin

{elliot.meyerson,olivier.francon,darren.sargent,babak,risto}@cognizant.com

## Abstract

Solving societal problems on a global scale requires the collection and processing of ideas and methods from diverse sets of international experts. As the number and diversity of human experts increase, so does the likelihood that elements in this collective knowledge can be combined and refined to discover novel and better solutions. However, it is difficult to identify, combine, and refine complementary information in an increasingly large and diverse knowledge base. This paper argues that artificial intelligence (AI) can play a crucial role in this process. An evolutionary AI framework, termed RHEA, fills this role by distilling knowledge from diverse models created by human experts into equivalent neural networks, which are then recombined and refined in a population-based search. The framework was implemented in a formal synthetic domain, demonstrating that it is transparent and systematic. It was then applied to the results of the XPRIZE Pandemic Response Challenge, in which over 100 teams of experts across 23 countries submitted models based on diverse methodologies to predict COVID-19 cases and suggest non-pharmaceutical intervention policies for 235 nations, states, and regions across the globe. Building upon this expert knowledge, by recombining and refining the 169 resulting policy suggestion models, RHEA discovered a broader and more effective set of policies than either AI or human experts alone, as evaluated based on real-world data. The results thus suggest that AI can play a crucial role in realizing the potential of human expertise in global problem-solving.

## 1 Introduction

Integrating knowledge and perspectives from a diverse set of experts is essential for developing better solutions to societal challenges, such as policies to curb an ongoing pandemic, slow down and reverse climate change, and improve sustainability [33, 41, 57, 63, 64]. Increased diversity in human teams can lead to improved decision-making [25, 62, 83], but as the scale of the problem and size of the team increases, it becomes difficult to discover the best combinations and refinements of available ideas [37]. This paper argues that artificial intelligence (AI) can play a crucial role in this process, making it possible to realize the full potential of diverse human expertise. Though there are many AI systems that take advantage of human expertise to improve automated decision-making [4, 31, 66], an approach to the general problem must meet a set of unique requirements: It must be able to incorporate expertise from diverse sources with disparate forms; it must be multi-objective since conflicting policy goals will need to be balanced; and the origins of final solutions must be traceable so that credit can be distributed back to humans based on their contributions. An evolutionary AI framework termed RHEA (for Realizing Human Expertise through AI) is developed in this paper to satisfy these requirements. Evolutionary AI, or population-based search, is a biologically-inspired method that often leads to surprising discoveries and insights [5, 15, 39, 48, 67]; it is also a natural fit here since the development of ideas in human teams mirrors an evolutionary process [14, 17, 38, 32]. Implementing RHEA for a particular application requires the following steps (Fig. 1):

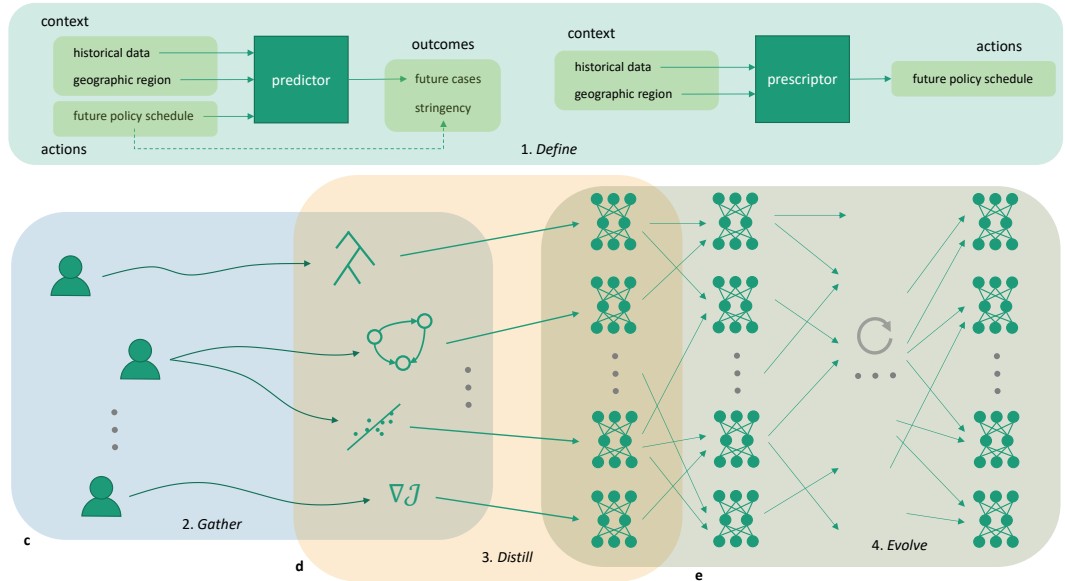

Figure 1: *The RHEA (Realizing Human Expertise through AI) framework.* The framework consists of four components: Defining the prediction and prescription tasks, gathering the human solutions, distilling them into a canonical form, and evolving the population of solutions further. **a,** The *predictor* maps context and actions to outcomes and thus constitutes a surrogate, or a "digital twin", of the real world. For example, in the Pandemic Response Challenge experiment, the context consisted of data about the geographic region for which the predictions were made, e.g., historical data of COVID-19 cases and intervention policies; actions were future schedules of intervention policies for the region; and outcomes were predicted future cases of COVID-19 along with the stringency of the policy. **b,** Given a predictor, the *prescriptor* generates actions that yield optimized outcomes across contexts. **c,** Humans are solicited to contribute expertise by submitting prescriptors using whatever methodology they prefer, such as decision rules, epidemiological models, classical statistical techniques, and gradient-based methods. **d,** Each submitted prescriptor is distilled into a canonical neural network that replicates its behavior. **e,** This population of neural networks is evolved further, i.e., the distilled models are recombined and refined in a parallelized, iterative search process. They build synergies and extend the ideas in the original solutions, resulting in policies that perform better than the original ones. For example, in the Pandemic Response Challenge, the policies recommend interventions that lead to minimal cases with minimal stringency.

1. **Define**. Define the problem in a formal manner so that solutions from diverse experts can be compared and combined.

2. **Gather**. Solicit and gather solutions from a diverse set of experts. Solicitation can take the form of an open call or a direct appeal to known experts.

3. **Distill**. Use machine learning to convert (distill) the internal structure of each gathered solution into a canonical form such as a neural network.

4. **Evolve**. Recombine and refine the distilled solutions using a population-based search to realize the complementary potential of the ideas in the expert-developed solutions.

RHEA is first illustrated through a formal synthetic example below, demonstrating how this process can result in improved decision-making. RHEA is then put to work in a large-scale international experiment on developing non-pharmaceutical interventions for the COVID-19 pandemic. The results show that broader and better policy strategies can be discovered in this manner, beyond those that would be available through AI or human experts alone. The results also highlight the value of soliciting diverse expertise, even if some of it does not have immediately obvious practical utility: AI may find ways to recombine it with other expertise to develop superior solutions.

To summarize, the main contributions of this paper are as follows: (1) Recognizing that bringing together diverse human expertise is a key challenge in solving many complex problems; (2) Identifying desiderata for an AI process that accomplishes this task; (3) Demonstrating that existing approaches do not satisfy these desiderata; (4) Formalizing a new framework, RHEA, to satisfy them; (5) Instantiating a first concrete implementation of RHEA using standard components; and (6) Evaluating this implementation in a global application: The XPRIZE Pandemic Response Challenge.

## 2  Illustrative Example

In this section, RHEA is applied to a formal synthetic setting where its principles and mechanics are transparent. It is thus possible to demonstrate how they can lead to improved results, providing a roadmap for when and how to apply it to real-world domains (see App. B for additional details).

Consider a policy-making scenario in which many new reasonable-sounding policy interventions are constantly being proposed, but there are high levels of nonlinear interaction between interventions and across contexts. Such interactions are a major reason why it is difficult to design effective policies and the main challenge that RHEA is designed to solve. They are unavoidable in complex real-world domains such as public health (e.g., between closing schools, requiring masks, or limiting international travel), traffic management (e.g., adding buses, free bus tokens, or bike lanes), and climate policy (e.g., competing legal definitions of "net-zero" or "green hydrogen", and environmental feedback loops) [19, 52, 60]. In such domains there exist diverse experts—e.g., policymakers, economists, scientists, local community leaders, and other stakeholders—whose input is worth soliciting before implementing interventions. In RHEA, this policy-making challenge can be formalized as follows:

**Define.** Suppose we are considering policy interventions $a_1, \ldots, a_n$. A policy action $A$ consists of some subset of these. Suppose we must be prepared to address contexts $c \in \{c_1, \ldots, c_m\}$, and we have a black-box predictor $\phi(c, A)$ to evaluate utility (Fig. 1a). In practice, $\phi$ will be a complex dynamical model such as an agent-based or neural-network-based predictor. In this example, to highlight the core behavior of RHEA, $\phi$ is a simple-to-define function containing the kinds of challenging nonlinearities we would like to address, such as context dependence, synergies, anti-synergies, threshold effects, and redundancy (the full utility function is detailed in Eq. 1). Similarly, $\psi$ is a simple cost function, defined as the total number of prescribed policy interventions. A prescriptor is a function $\pi(c) = A$ (Fig. 1b). The goal is to find a Pareto front of prescriptors across the outcomes of utility $\phi$ and cost $\psi$. Note that the search space is vast: There are $2^{mn}$ possible prescriptors.

**Gather.** Suppose prescriptors of unknown functional form have been gathered (Fig. 1c) from three experts: one "generalist", providing general knowledge that applies across contexts (see Fig. 2c for an example); and two "specialists", providing knowledge that is of higher quality (i.e. lower cost-per-utility) but applies only to a few specific contexts (Fig. 2a-b).

**Distill.** Datasets for distillation can be generated by running each expert prescriptor over all contexts. The complete behavior of a prescriptor can then be visualized as a binary grid, where a black cell indicates the inclusion of an intervention in the prescription for a given context (Fig. 2a-c). This data can be used to convert the expert prescriptors into rule sets or neural networks (Fig. 1d, App. B.2).

**Evolve.** These distilled models can then be injected into an initial population and evolved using multi-objective optimization [16] (Fig. 1e). The full optimal Pareto front is obtained as a result.

With this formalization, it is possible to construct a synthetic example of RHEA in action, as shown in Fig. 2. It illustrates the optimal Pareto front. Importantly, this front is discoverable by RHEA, but not by previous machine learning techniques such as Mixture-of-Experts (MoE) [42] or Weighted Ensembles [18], or by the experts alone. RHEA is able to recombine the internal structure of experts across contexts (e.g., by adding $a_3, a_4, a_5$ to $a_1, a_2$ in $c_1$). It can innovate beyond the experts by adding newly-applicable interventions ($a_6$). It can also refine the results by removing interventions that are now redundant or detrimental ($a_5$ in $c_2$), and by mixing in generalist knowledge. In contrast, the discoveries of MoE are restricted to mixing expert behavior independently at each context, and Weighted Ensemble solutions can only choose a single combination of experts to apply everywhere.

The domain also illustrates why it is important to utilize expert knowledge in the first place. The high-dimensional solution space makes it very difficult for evolution alone (i.e. not starting from distilled expert prescriptors) to find high-quality solutions, akin to finding needles in a haystack. Experimental results confirm that RHEA discovers the entire optimal Pareto front reliably, even

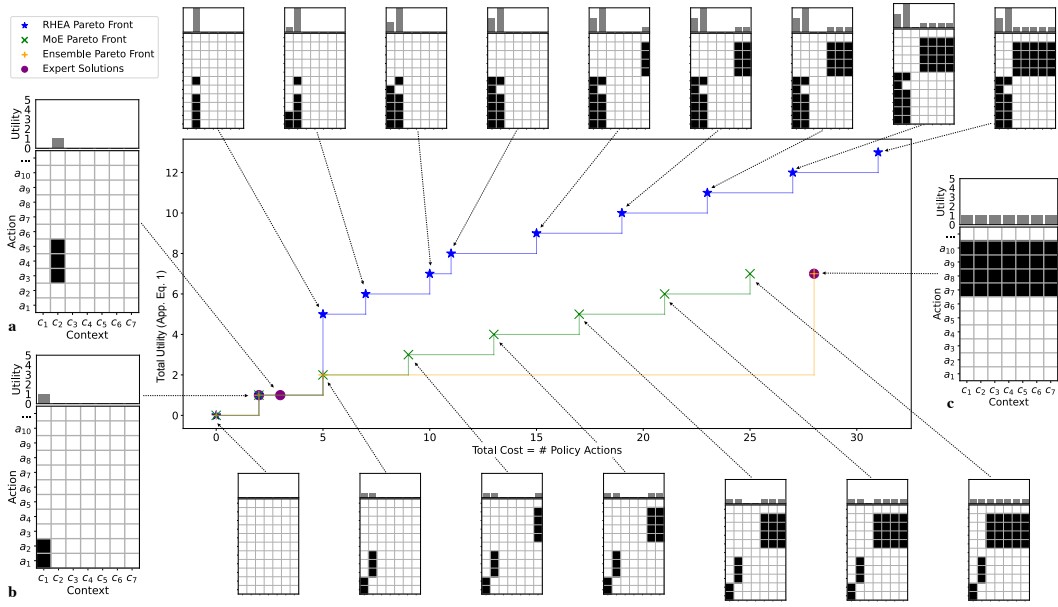

Figure 2: *An Illustration of RHEA in a Synthetic Domain.* The plots show the Pareto front of prescriptors discovered by RHEA vs. those of alternative prescriptor combination methods, highlighting the kinds of opportunities RHEA is able to exploit. The specialist expert prescriptors **a** and **b** and the generalist expert prescriptor **c** are useful but suboptimal on their own (purple ●'s). RHEA recombines and innovates upon their internal structure and is able to discover the full optimal Pareto front (blue ⋆'s). This front dominates that of Mixture-of-Experts (MoE; green ×'s), which can only mix expert behavior independently in each context. It also dominates that of Weighted Ensembling (yellow +'s), which can only choose a single combination of experts to apply everywhere. Evolution alone (without expert knowledge) also struggles in this domain due to the vast search space (App. Fig. 6), as do MORL methods (App. Fig. 7,8). Thus, RHEA unlocks the latent potential in expert solutions.

as the number of available interventions increases, while evolution alone does not (App. Fig. 6). Multi-objective reinforcement learning (MORL) methods also struggle in this domain (App. Fig. 7,8). Thus, RHEA harnesses the latent potential of expert solutions. It uses pieces of them as building blocks and combines them with novel elements to take full advantage of them. This ability can be instrumental in designing effective policies for complex real-world tasks. Next, RHEA is put to work on one particularly vexing task: optimizing pandemic intervention policies.

## 3   The XPRIZE Pandemic Response Challenge

The XPRIZE Pandemic Response Challenge [10, 11] presented an ideal opportunity for demonstrating the RHEA framework. XPRIZE is an organization that conducts global competitions, fueled by large cash prizes, to motivate the development of underfunded technologies. Current competitions target wildfires, desalination, carbon removal, meat alternatives, and healthy aging [81]. In 2020 and 2021, the XPRIZE Pandemic Response Challenge was designed and conducted [78], challenging participants to develop models to suggest optimal policy solutions spanning the tradeoff between minimizing new COVID-19 cases and minimizing the cost of implemented policy interventions.

**Define.** The formal problem definition was derived from the Oxford COVID-19 government response tracker dataset [27, 54, 74], which was updated daily from March 2020 through December 2022. This dataset reports government intervention policies ("IPs") on a daily basis, following a standardized classification of policies and corresponding ordinal stringency levels in $\mathbb{Z}_5$ (used to define IP "cost") to enable comparison across geographical regions ("geos"), which include nations and subnational regions such as states and provinces. The XPRIZE Challenge focused on 235 geos (App. Fig 9) and those 12 IPs over which governments have immediate daily control [54]: school closings, workplace closings, cancellation of public events, restrictions on gathering size, closing of public transport,

stay at home requirements, restrictions on internal movement, restrictions on international travel, public information campaigns, testing policy, contact tracing, and facial covering policy. Submissions for Phase 1 were required to include a runnable program ("predictor") that outputs predicted cases given a geo, time frame, and IPs through that time frame (Fig. 1a). Submissions for Phase 2 were required to include a set of runnable programs ("prescriptors"), which, given a geo, time frame, and relative IP costs, output a suggested schedule of IPs ("prescription") for that geo and time frame (Fig. 1b). By providing a set of prescriptors, submissions could cover the tradeoff space between minimizing the cost of implementing IPs and the expected number of new cases. Since decision makers for a particular geo could not simultaneously implement multiple prescriptions from multiple teams, prescriptions were evaluated not in the real world but with a predictor $\phi$ (from Phase 1), which forecasts how case numbers change as a result of a prescription. The formal problem definition, requirements, API, and code utilities are publicly available [10]. Teams were encouraged to incorporate specialized knowledge in geos with which they were most familiar. The current study focuses on the prescriptors created in Phase 2. There are $\approx 10^{620}$ possible schedules for a *single geo* for 90 days, so brute-force search is not an option. To perform well, prescriptors must implement principled ideas to capture domain-specific knowledge about the structure of the pandemic.

**Gather.** Altogether, 102 teams of experts from 23 countries participated in the challenge. Some teams were actively working with local governments to inform policy [49, 53]; other organizations served as challenge partners, including the United Nations ITU and the City of Los Angeles [80]. The set of participants was diverse, including epidemiologists, public health experts, policy experts, machine learning experts, and data scientists. Consequently, submissions took advantage of diverse methodologies, including epidemiological models, decision rules, classical statistical methods, gradient-based optimization, various machine learning methods, and evolutionary algorithms, and exploited various auxiliary data sources to get enhanced views into the dynamics of particular geos [79] (Fig. 1c). The Phase 2 evaluations showed substantial specialization to different geos for different teams, a strong indication that there was diversity that could be harnessed. Many submissions also showed remarkable improvement over strong heuristic baselines, indicating that high-quality expertise had been gathered successfully. Detailed results of the competition are publicly available [11]; this study focuses on the ideas in them in the aggregate.

**Distill.** A total of 169 prescriptors were submitted to the XPRIZE Challenge. After the competition, for each of these gathered prescriptors $\pi_i$, an autoregressive neural network (NN) $\hat{\pi}_i$ with learnable parameters $\theta_i$ was trained with gradient descent to mimic its behavior, i.e. to distill it [30, 31] (Fig. 1d; App. C.1). Each NN was trained on a dataset of 212,400 input-output pairs, constructed by querying the corresponding prescriptor $n_q$ times, i.e., through behavioral cloning:

$$\theta_i^* = \operatorname*{argmin}_{\theta_i} \int_{\mathcal{Q}} p(q) \left\| \pi_i(q) - \hat{\pi}_i\Big(\kappa(q, \pi_i(q), \phi); \theta_i\Big) \right\|_1 dq \tag{1}$$

$$\approx \operatorname*{argmin}_{\theta_i} \frac{1}{n_q} \sum_{j=1}^{n_q} \left\| \pi_i(q_j) - \hat{\pi}_i\Big(\kappa(q, \pi_i(q_j), \phi); \theta_i\Big) \right\|_1, \tag{2}$$

where $q \in \mathcal{Q}$ is a query and $\kappa$ is a function that maps queries (specified via the API in Define) to input data, i.e., *contexts*, with a canonical form. Each (date range, geo) pair defines a query $q$, with $\pi_i(q) \in \mathbb{Z}_5^{90 \times 12}$ the policy generated by $\pi_i$ for this geo and date range, and $\phi(q, \pi_i(q)) \in \mathbb{R}^{90}$ the predicted (normalized) daily new cases. Distilled models were implemented in Keras [7] and trained with Adam [35] using L1 loss (since policy actions were on an ordinal scale) (see App. C.1).

**Evolve.** These 169 distilled models were then placed in the initial population of an evolutionary AI process (Fig. 1e). This process was based on the same Evolutionary Surrogate-assisted Prescription (ESP) method [24] previously used to evolve COVID-19 IP prescriptors from scratch [50]. In standard ESP, the initial population (i.e., before any evolution takes place) consists only of NNs with randomly generated weights. By replacing random neural networks with the distilled neural networks, ESP starts from diverse high-quality expert-based solutions, instead of low-quality random ones. ESP can then be run as usual from this starting point, recombining and refining solutions over a series of generations to find better tradeoffs between stringency and cases, using Pareto-based multi-objective optimization [16] (App. C.2). Providing a Pareto front of policy strategy options is critical, because most decision-makers will not simply choose the most extreme strategies (i.e. IPs with maximum stringency, or no IPs at all), but are likely to choose a tradeoff point appropriate for their particular political, social and economic scenario (Fig. 3d shows the real-world distribution of IP stringencies).

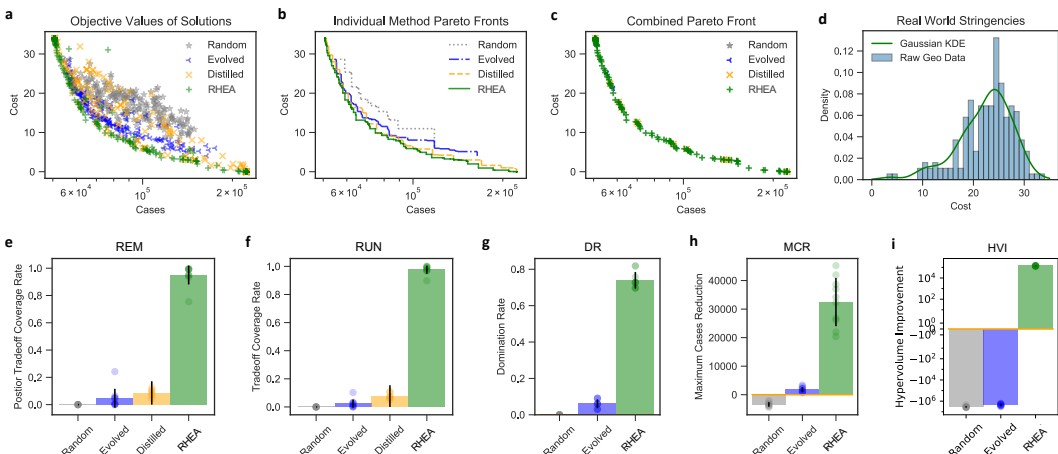

Figure 3: *Quantitative comparison of solutions.* **a,** Objective values for all solutions in the final population of a single representative run of each method. **b,** Pareto curves for these runs. Distilled provides improved tradeoffs over Random and Evolved (from random), and RHEA pushes the front out beyond Distilled. **c,** Overall Pareto front of the union of the solutions from these runs. The vast majority of these solutions are from RHEA. **d,** The distribution of actual stringencies implemented in the real world across all geos at the prescription start date, indicating which Pareto solutions real-world decision makers would likely select, i.e., which tradeoffs they prefer. **e,** Given this distribution, the proportion of the time the solution selected by a user would be from a particular method (the REM metric); almost all of them would be from RHEA. **f,** The same metric, but based on a uniform distribution of tradeoff preference (RUN) **g,** Domination rate (DR) w.r.t. Distilled, i.e. how much of the Distilled Pareto front is strictly dominated by another method's front. While Evolved (from scratch) sometimes discovers better solutions than those distilled from expert designs, RHEA improves ≈75% of them. **h,** Max reduction of cases (MCR) compared to Distilled across all stringency levels. **i,** Dominated hypervolume improvement (HVI) compared to Distilled. For each metric, RHEA substantially outperforms the alternatives, demonstrating that it creates improved solutions over human and AI design, and that those solutions would likely be preferred by human decision-makers. (Bars show mean and st.dev. See App. C.3 for technical details of each metric.)

Evolution from the distilled models was run for 100 generations in 10 independent trials to produce the final RHEA models. As a baseline, evolution was run similarly from scratch. As a second baseline, RHEA was compared to the full set of distilled models. A third baseline was models with randomly initialized weights, which is often a meaningful starting point in NN-based policy search [68]. All prescriptor evaluations, including those during evolution, were performed using the same reference predictor as in the XPRIZE Challenge itself; this predictor was evaluated in depth in prior work [50].

*Results.* The performance results are shown in Fig. 3. As is clear from the Pareto plots (Fig. 3a-c) and across a range of metrics (Fig. 3e-i), the distilled models outperform the random initial models, thus confirming the value of human insight and the efficacy of the distillation process. Evolution then improves performance substantially from both initializations, with distilled models leading to the best solutions. Thus, the conclusions of the illustrative example are substantiated in this real-world domain: RHEA is able to leverage knowledge contained in human-developed models to discover solutions beyond those from the AI alone or humans alone. The most critical performance metric is the empirical R1-metric (REM; [28]), which estimates the percentage of time a decision-maker with a fixed stringency budget would choose a prescriptor from a given approach among those from all approaches. For RHEA, REM is nearly 100%. In other words, not only does RHEA discover policies that perform better, but they are also policies that decision-makers would be likely to adopt.

## 4   Characterizing the Innovations

Two further sets of analyses characterize the RHEA solutions and the process of discovering them. First, IP schedules generated for each geo by different sets of policies were projected to 2D via

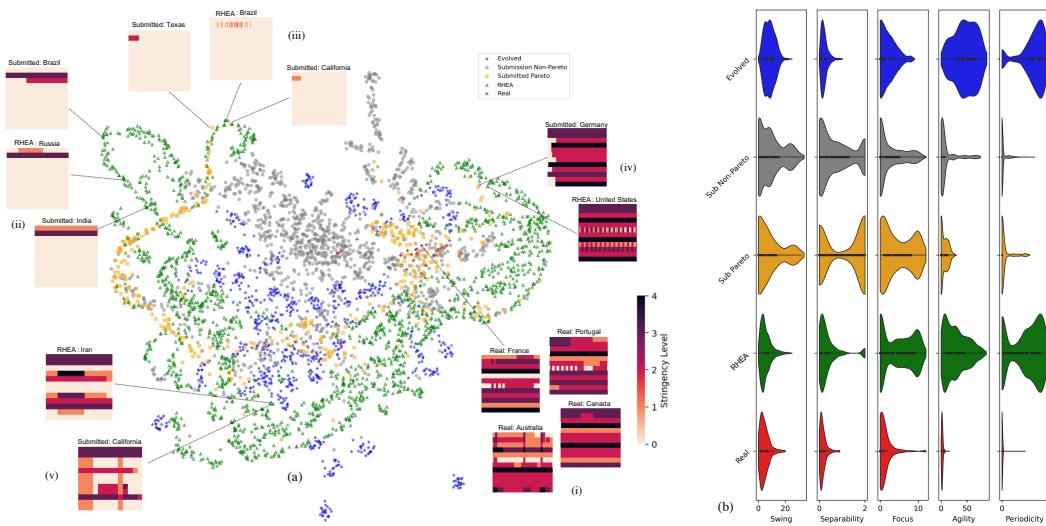

Figure 4: *Dynamics of IP schedules discovered by RHEA.* **a,** UMAP projection of geo IP schedules generated by the policies (App. C.4). The schedules from high-performing submitted expert models are concentrated around a 1-dimensional manifold organized by overall cost (seen as a yellow arc). This manifold provides a scaffolding upon which RHEA elaborates, interpolates, and expands. Evolved policies, on the other hand, are scattered more discordantly (seen as blue clusters), ungrounded by the experts. **b,** To characterize how RHEA expands upon this scaffolding, five high-level properties of IP schedules were identified and their distributions were plotted across the schedules. For each, RHEA finds a balance between the grounding of expert submissions (i.e., regularization) and their recombination and elaboration (i.e., innovation), though this balance manifests in distinct ways. For swing and separability, RHEA is similar to real schedules, but finds that the high separability proposed by some expert models can sometimes be useful. RHEA finds the high focus of the expert models even more attractive; in practice, they could provide policy-makers with simpler and clearer messages about how to control the pandemic. For focus, agility, and periodicity, RHEA pushes beyond areas explored by the submissions, finding solutions that humans may miss. The example schedules shown in **a(i-v)** illustrate these principles in practice (rows are IPs sorted from top to bottom as listed in Sec. 3; column are days in the 90-day period; darker color means more stringent). **(i)** Real-world examples demonstrate that although agility and periodicity require some effort to implement, they have occasionally been utilized (e.g. in Portugal and France); **(ii)** a simple example of how RHEA generates useful interpolations of submitted non-Pareto schedules, demonstrating how it realizes latent potential even in some low-performing solutions, far from schedules evolved from scratch; **(iii)** another useful interpolation, but achieved via higher agility than Pareto submissions; **(iv)** a high-stringency RHEA schedule that trades swing and separability for agility and periodicity compared to its submitted neighbor; and **(v)** a medium-stringency RHEA schedule with lower swing and separability and higher focus than its submitted neighbor. Overall, these analyses show how RHEA realizes the latent potential of the raw material provided by the human-created submissions.

UMAP [45] to visualize the distribution of their behavior (Fig. 4a). Note that the schedules from the highest-performing (Pareto) submitted policies form a continuous 1D manifold across this space, indicating continuity of tradeoffs. This manifold serves as scaffolding upon which RHEA recombines, refines, and innovates; these processes are the same as in the illustrative example, only more complex. Evolution alone, on the other hand, produces a discordant scattering of schedules, reflecting its unconstrained exploratory nature, which is disadvantageous in this domain. What kind of structure does RHEA harness to move beyond the existing policies? Five high-level properties were identified that characterize how RHEA draws on submitted models in this domain: *swing* measures the stringency difference between the strictest and least strict day of the schedule; *separability* measures to what extent the schedule can be separated into two contiguous phases of different stringency levels; *focus* is inversely proportional to the number of IPs used; *agility* measures how often IPs change; and *periodicity* measures how much of the agility can be explained by weekly periodicity

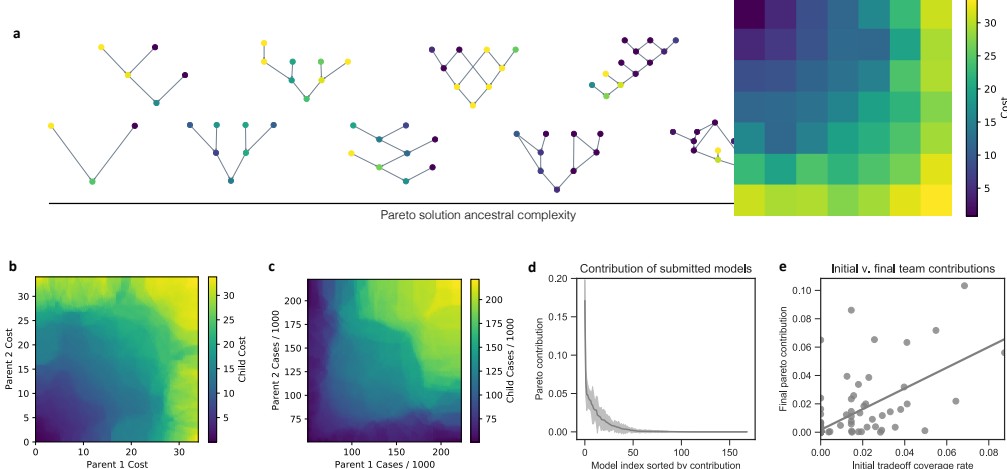

Figure 5: *Dynamics of evolutionary discovery process.* **a,** Sample ancestries of prescriptors on the RHEA Pareto front. Leaf nodes are initial distilled models; the final solutions are the root. The history of recombinations leading to different solutions varies widely in terms of complexity, with apparent motifs and symmetries. The ancestries show that the search is behaving as expected, in that the cost of the child usually lands between the costs of its parents (indicated by color). This property is also visualized in **b** (and **c**), where child costs (and cases) are plotted over all recombinations from all trials (k-NN regression, k = 100). **d,** From ancestries, one can compute the relative contribution of each expert model to the final RHEA Pareto front (App C.5). This contribution is remarkably consistent across the independent runs, indicating that the approach is reliable (mean and st.dev. shown). **e,** Although there is a correlation between the performance of teams of expert models and their contribution to the final front, there are some teams with unimpressive quantitative performance in their submissions who end up making outsized contributions through the evolutionary process. This result highlights the value of soliciting a broad diversity of expertise, even if some of it does not have immediately obvious practical utility. AI can play a role in realizing this latent potential.

(Fig. 4b; App. C.4). Some ideas from submitted policies, e.g., increased separability and focus, are readily incorporated into RHEA policies. Others, e.g. increased focus, agility, and periodicity, RHEA is able to utilize beyond the range of policies explored by the human designs. The examples in Fig. 4a illustrate these properties in practice. Example (i) shows a number of real policies, suggesting that geos are capable of implementing diverse and innovative schedules similar to those discovered by RHEA; e.g., weekly periodicity was actually implemented for a time in Portugal and France. Examples (ii-v) show RHEA schedules and their nearest submitted neighbors, demonstrating how innovations can manifest as interpolations or extrapolations of submitted policies. For instance, one opportunity is to focus on a smaller set of IPs; another is to utilize greater agility and periodicity. This analysis shows how RHEA can lead to insights on where improvements are possible.

Second, to understand how RHEA discovered these innovations, an evolutionary history can be reconstructed for each solution, tracing it back to its initial distilled ancestors (Fig. 5). Some final solutions stem from a single beneficial crossover of distilled parents, while others rely on more complex combinations of knowledge from many ancestors (Fig. 5a). While the solutions are more complex, the evolutionary process is similar to that of the illustrative example: It proceeds in a principled manner, with child models often falling between their parents along the case-stringency tradeoff (Fig. 5b-c). Based on these evolutionary histories, one can compute the relative contribution of each expert model to the final RHEA Pareto front (App C.5). These contributions are highly consistent across independent runs, indicating that the approach is reliable (Fig. 5d). Indeed in the XPRIZE competition, this contribution amount was used as one of the quantitative metrics of solution quality [12]. Remarkably, although there is a correlation between the performance of expert models and their contribution to the final front, there are also models that do not perform particularly well, but end up making outsized contributions through the evolutionary process (Fig. 5e; see also Fig. 4a(ii)). This result highlights the value of soliciting a broad diversity of expertise, even if some of it does not have immediately obvious practical utility. AI can play a role in realizing this latent potential.

## 5 Discussion

**Alternative Policy Discovery Methods.**   Our implementation of RHEA uses established methods in both the Distill and Evolve steps; the technical novelty comes from their unique combination in RHEA to unlock diverse human expertise. Popular prior methods for combining diverse models include ensembling [18] and Mixture-of-Experts [42], but, as highlighted in Fig. 2, although multi-objective variants have been explored in prior work [36], neither of these methods can innovate beyond the scaffolding provided by the initial experts. Evolution is naturally suited for this task: Crossover is a powerful way to recombine expert models, mutation allows innovating beyond them, and population-based search naturally supports multiobjective optimization. Other approaches for policy optimization include contextual bandits [73], planning-based methods [66], and reinforcement learning [29, 69], and an interesting question is how they might play a role in such a system. One approach could be to use evolutionary search for recombination and use another method for local improvement, akin to hybrid approaches used in other settings [6] (See App. A for a longer discussion).

**Theory.**   It is intuitive why expert knowledge improves RHEA's search capability. However, any theoretical convergence analysis will depend on the particular implementation of RHEA. The present implementation uses NSGA-II, the convergence of which has recently been shown to depend critically on the size of jumps in the optimization landscape, i.e. roughly the maximum size of non-convex regions [20, 21]. On the ONEJUMPZEROJUMP benchmark, the tightest known upper-bound for convergence to the full ground truth Pareto front is $O(N^2 n^k / \Theta(k)^k)$, where $k$ is a measure of the jump size, $n$ is the problem dimensionality, and $N$ is the (sufficiently large) population size. In other words, a smaller jump size leads to a drastic convergence speed up. Distilling useful, diverse experts is conceptually analogous to decreasing the jump size. This effect is apparent in the illustrative domain, where the experts provide building blocks that can be immediately recombined to discover better solutions, but that are difficult to discover from scratch (Fig. 2). This interpretation is borne out in the experiments: RHEA continues to converge quickly as the action space (i.e. problem dimensionality) increases, whereas evolution regresses to only being able to discover the most convex (easily-discoverable) portions of the Pareto front (App. Fig. 6).

**Generalizability.**   RHEA can be applied effectively to policy-discovery domains where (1) the problem can be formalized with contexts, actions, and outcomes, (2) there exist diverse experts from which solutions can be gathered, and (3) the problem is sufficiently challenging. In contrast, RHEA would not be effective, (1) if the problem is too easy, so that the input from human experts would not be necessary, (2) if the problem is hard, but no useful and diverse experts exist, and (3) if there is no clear way to define context and/or action variables upon which the experts agree.

The modularity of RHEA allows different implementations of components to be designed for different domains, such as those related to sustainability, engineering design, and public health. One particularly exciting opportunity for RHEA is climate policy, which often includes complex interactions between multiple factors [46]. For example, given the context of the current state of the US energy grid and energy markets, the green hydrogen production subsidies introduced by the Inflation Reduction Act will in fact lead to *increases* in carbon emissions, *unless* the Treasury Department enacts three distinct regulations in the definition of "green hydrogen" [60]. It is precisely this kind of policy combination that RHEA could help discover, and such a discovery process could be an essential part of a climate policy application. For example, the En-Roads climate simulator supports diverse actions across energy policy, technology, and investment, contexts based on social, economic, and environmental trajectories, and multiple competing outcomes, including global temperature, cost of energy, and sea-level rise [8]. Users craft policies based on their unique priorities and expertise. RHEA could be used with a predictor like En-Roads to discover optimized combinations of expert climate policies that trade-off across temperature change and other the outcomes that users care about most.

**Ethics and Broader Impact.**   As part of the UN AI for Good Initiative, we are currently building a platform for formalizing and soliciting expert solutions to SDG goals more broadly [55]. Ethical considerations when deploying such systems are outlined below. See App. D for further discussion.

*Fairness.* In such problems with diverse stakeholders, breaking down costs and benefits by affected populations and allowing users to input explicit constraints to prescriptors can be crucial for generating feasible and equitable models. In this platform, RHEA could take advantage of knowledge that local experts provide and learn to generalize it; by treating each contributed model as a black box,

it is agnostic to the type of models used, thus helping to make the platform future-proof. Fairness constraints can also be directly included in RHEA's multiple objectives.

*Governance and Democratic Accountability.* An important barrier in the adoption of AI by real-world decision-makers is trust [44, 65]. For example, such systems could be used to justify the decisions of bad actors. RHEA provides an advantage here: If the initial human-developed models it uses are explainable (e.g. are based on rules or decision trees), then a user can trust that suggestions generated by RHEA models are based on sensible principles, and can trace and interrogate their origins. Even when the original models are opaque, trust can be built by extracting interpretable rules that describe prescriptor behavior, which is feasible when the prescriptors are relatively compact and shallow [71, 72], as in the experiments in this paper. That is, RHEA models can be effectively audited—a critical property for AI systems maintained by governments and other policy-building organizations.

*Data Privacy and Security.* Since experts submit complete prescriptors, no sensitive data they may have used to build their prescriptors needs to be shared. In the Gather step in Sec. 3, each expert team had an independent node to submit their prescriptors. The data for the team was generated by running their prescriptors on their node. The format of the data was then automatically verified, to ensure that it complied with the Defined API. Verified data from all teams was then aggregated for the Distill & Evolve steps. Since the aggregated data must fit an API that does not allow for extra data to be disclosed, the chance of disclosing sensitive data in the Gather phase is minimized.

*External Oversight.* Although the above mechanisms could all yield meaningful steps in addressing a broad range of ethical concerns, they cannot completely solve all issues of ethical deployment. So, it is critical that the system is not deployed in an isolated way, but integrated into existing democratic decision-making processes, with appropriate external oversight. Any plan for deployment should include a disclosure of these risks to weigh against the potential societal benefits.

*Sustainability and Accessibility.* Due to the relatively compact model size, RHEA uses orders of magnitude less compute and energy than many other current AI systems, which is critical for creating uptake by decision-makers around the world who do not have access to extensive computational resources or for whom energy usage is becoming an increasingly central operational consideration.

**Limitations.** Understanding the limitations of the presented RHEA implementation is critical for establishing directions for future work. The cost measure used in this paper was uniform over IPs, an unbiased way to demonstrate the technology, but, for a prescriptor to be used in a particular geo, costs of different IPs should be calibrated based on geo-specific cost-analysis. The geo may also have some temporal discounting in its cases and cost objectives. For consistency with the XPRIZE, they were not included in the experiments in this paper but can be naturally incorporated into RHEA in the future. When applying surrogate-developed policies to the real world, approximation errors can compound over time. Thus, user-facing applications of RHEA could benefit from the inclusion of uncertainty measures [26, 58], inverse reinforcement learning [2, 70], as well as humans-in-the-loop to prevent glaring errors. Distillation could also be limited in cases where expert models use external data sources with resulting effects not readily approximated by the inputs specified in the defined API. If this were an issue in future applications, it could be addressed by training models that generalize across domain spaces [47, 59]. RHEA prescriptors were evaluated in the same surrogate setting as prescriptors in the XPRIZE, but not yet in hands-on user studies. Hands-on user evaluation is a critical step but requires a completely different kind of research effort, i.e. one that is political and civil, rather than computational. Our hope is that the publication of the results of RHEA makes the real-world incorporation of these kinds of AI decision-assistants more likely.

**Conclusion.** This paper motivated, designed, and evaluated a framework called RHEA for bringing together diverse human expertise systematically to solve complex problems. The promise of RHEA was illustrated with an initial implementation and an example application; it can be extended to other domains in future work. The hope is that, as a general and accessible system that incorporates input from diverse human sources, RHEA will help bridge the gap between human-only decision-making and AI-from-data-only approaches. As a result, decision-makers can start adopting powerful AI decision-support systems, taking advantage of the latent real-world possibilities such technologies illuminate. More broadly, the untapped value of human expertise spread across the world is immense. Human experts should be actively encouraged to continually generate diverse creative ideas and contribute them to collective pools of knowledge. This study shows that AI has a role to play in realizing the full value of this knowledge, thus serving as a catalyst for global problem-solving.

## Acknowledgements

We would like to thank XPRIZE for their work in instigating, developing, publicizing, and administering the Pandemic Response Challenge, as well as the rest of the Cognizant AI Labs research group for their feedback on experiments and analysis. We would also like to thank Conor Hayes for advice on running the MORL comparisons, and Benjamin Doerr for advice on NSGA-II theory.

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

# Appendix

## A   Related Work

The RHEA method builds on a long tradition of leveraging diversity in machine learning, as well as methods for policy discovery in general.

### A.1   Harnessing diversity in AI

Machine learning (ML) models generally benefit from diversity in the data on which they are trained [34]. At a higher level, it has long been known that diverse models for a single task may be usefully combined to improve performance on the task. Methods for such combination usually fall under the label of *ensembling* [18]. By far, the most popular ensembling method is to use a linear combination of models. Mixture-of-Experts (MoE) approaches use a more sophisticated approach of conditionally selecting which models to use based on the input [42]. However, as highlighted in Fig. 2, although some multi-objective variants have been explored in prior work [36], neither of these methods are inherently sufficient for the kind of policy discovery required by RHEA. In particular, such methods are not multi-objective, and provide no method of innovating beyond the scaffolding provided by the individual experts.

An orthogonal approach to harnessing diversity within a single task is to exploit regularities across multiple tasks, by learning more than one task in a single model [85]. In the extreme case, a single model may be trained across many superficially unrelated tasks with the goal of learning shared structure underlying the problem-solving universe [47, 59]. In this paper, it was possible to specialize the expert models to different regions and pandemic states, but the input-output spaces of the models were uniform to enable a consistent API. Future work could generalize RHEA to cases where the expert models are trained on different, but related, problems that could potentially benefit from one another.

Finally, there is a rich history of managing and exploiting diverse solutions in evolutionary algorithms: from early work on preserving diversity to prevent premature convergence [43] and well-established work on multi-objective optimization [16], to more recent research on novelty search and diversity for diversity's sake [40], and to the burgeoning field of Quality Diversity, where the goal is to discover high-performing solutions across an array of behavioral dimensions [56]. RHEA is different from these existing methods because it is not about discovering diverse solutions *de novo*, but rather about harnessing the potential of diverse human-created solutions. Nonetheless, the scope and success of such prior research illustrate why evolutionary optimization is well-suited for recombining and innovating upon diverse solutions.

### A.2   Alternative approaches to policy discovery

In this paper, evolutionary optimization was used as a discovery method because it is most naturally suited for this task: crossover is a powerful way to recombine expert models, mutation allows innovating beyond them, and population-based search naturally supports multiobjective optimization. Other approaches for policy optimization include contextual bandits [73], planning-based methods [66], and reinforcement learning [69], and an interesting question is whether they could be used in this role as well.

Although less common than in evolutionary optimization, multi-objective approaches have been developed for such methods [22, 82]. However, because they aim at improving a single solution rather than a population of solutions, they tend to result in less exploration and novelty than evolutionary approaches [51]. One approach could be to use evolutionary search for recombination and use one of these non-evolutionary methods for local improvement. Such hybrid approaches have been used in other settings [6], and would be an interesting avenue of future work with RHEA.

## B   Illustrative Example

This section details the methods used in the formal synthetic example.

## B.1 Definition of Utility Function

The utility predictor $\phi$ is defined to be compact and interpretable, while containing the kinds of nonlinearities leading to optimization challenges that RHEA is designed to address:

$$\phi(c, A) = \begin{cases} 1, & \text{if } c = c_1 \wedge A = \{a_1, a_2\} \\ 2, & \text{if } c = c_1 \wedge A = \{a_1, a_2, a_3, a_4, a_5\} \\ 3, & \text{if } c = c_1 \wedge A = \{a_1, a_2, a_3, a_4, a_5, a_6\} \\ 4, & \text{if } c = c_2 \wedge A = \{a_1, a_2, a_3, a_4, a_5, a_6\} \\ 5, & \text{if } c = c_2 \wedge A = \{a_1, a_2, a_3, a_4, a_6\} \\ 1, & \text{if } c = c_2 \wedge A = \{a_3, a_4, a_5\} \\ 1, & \text{if } A = \{a_7, a_8, a_9, a_{10}\} \\ 0, & \text{otherwise.} \end{cases} \tag{1}$$

In this definition, the non-zero-utility cases represent context-dependent *synergies* between policy interventions; they also represent *threshold effects* where utility is only unlocked once enough of the useful interventions are implemented. The interventions that are not present in these cases yield *anti-synergies*, i.e. they negate any positive policy effects. The contexts $c_1$ and $c_2$ represent *similar but distinct contexts* in which similar but distinct combinations of interventions are useful and can inform one another. In $c_2$, $a_5$ becomes *redundant* once $a_6$ is included.

## B.2 Analytic Distillation

Since the context and action spaces are discrete in this domain, prescriptors can be analytically distilled based on the dataset describing their full behavior (i.e., the binary grids in Fig. 2). For example, these prescriptors can be distilled into rule-based or neural network-based prescriptors.

Consider rule-based prescriptors of the form: $\pi = [C_1 \mapsto A_1, \ldots, C_r \mapsto A_r]$, where $C_i \subseteq \{c_1, \ldots, c_m\}$ and $A_i \subseteq \{a_1, \ldots, a_n\}$ are subsets of the possible contexts and policy interventions, respectively. These prescriptors have a variable number of rules $r \geq 0$. Given a context $c$, $\pi(c)$ prescribes the first action $A_i$ such that $c \in C_i$, and prescribes the empty action $A_o = \emptyset$ if no $C_i$ contains $c$. Then, the gathered expert prescriptors with behavior depicted in Fig. 2a-c can be compactly distilled as $\pi_1 = [\{c_1\} \mapsto \{a_1, a_2\}]$, $\pi_2 = [\{c_2\} \mapsto \{a_3, a_4, a_5\}]$, and $\pi_3 = [\{c_1, c_2, c_3, c_4, c_5, c_6, c_7\} \mapsto \{a_7, a_8, a_9, a_{10}\}]$, respectively.

Similarly, consider neural-network-based prescriptors with input nodes $c_1, \ldots, c_m$, output nodes $a_1, \ldots, a_n$, and hidden nodes with ReLU activation and no bias. For every unique action $A_i$ prescribed by a prescriptor $\pi$, let $C_i$ be the set of contexts $c$ for which $\pi(c) = A_i$. Add a hidden node $h_i$ connected to each input $c \in C_i$ and each output $a \in A_i$. Let all edges have weight one. When using this model, include a policy intervention $a_i$ in the prescribed action if its activation is positive. Then, distilled versions of the expert prescriptors can be compactly described by their sets of directed edges: $\pi_1 = \{(c_1, h_1), (h_1, a_1), (h_1, a_2)\}$, $\pi_2 = \{(c_2, h_1), (h_1, a_3), (h_1, a_4), (h_1, a_5)\}$, and $\pi_3 = \{(c_1, h_1), (c_2, h_1), \ldots, (c_7, h_1), (h_1, a_7), (h_1, a_8), (h_1, a_9), (h_1, a_{10})\}$.

Both rules and neural networks provide a distilled prescriptor representation amenable to evolutionary optimization.

## B.3 Evolution of Analytically Distilled Models

For experimental verification, the distilled rule-set models were used to initialize a minimal multi-objective evolutionary AI process. This process was built from standard components including a method for recombination and variation of rule sets, non-dominated sorting [16], duplicate removal, and truncation selection. In the RHEA setup, the distilled versions of the gathered expert prescriptors were used to initialize the population and were reintroduced every generation. In the evolution alone setup, all instances of distilled models were replaced with random ones. The Python code for running these experiments can be found at `https://github.com/cognizant-ai-labs/rhea-demo`.

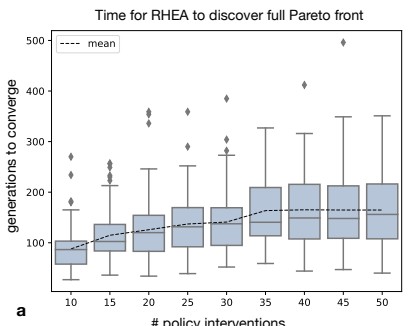
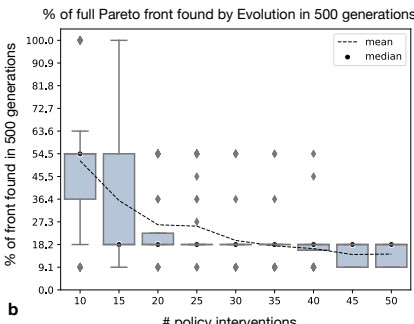

Figure 6: *Experimental results comparing RHEA vs. Evolution alone (i.e., without knowledge of gathered expert solutions) in the illustrative domain.* Whiskers show 1.5×IQR; the middle bar is the median. **a,** RHEA exploits latent expert knowledge to reliably and efficiently discover the full optimal Pareto front, even as the number of available policy interventions $n$ increases (there are $2^n$ possible actions for each context; 100 trials each). **b,** Evolution alone does not reliably discover the front even with 10 available interventions, and its performance drops sharply as the number increases (100 trials each). Thus, diverse expert knowledge is key to discovering optimal policies.

## B.4  Comparison to multi-objective reinforcement learning

Multi-objective reinforcement learning (MORL) is a growing area of research that aims to deploy the recent successes of reinforcement learning (RL) to multi-objective domains [29]. A natural question is: Is RHEA needed, or can MORL methods be directly applied from scratch (without expert knowledge) and reach similar or better performance?

To answer this question, comparisons were performed with a suite of state-of-the-art MORL techniques [23] in the Illustrative domain. Preliminary tests were run with several of the recent algorithms, namely, GPI-LS [1], GPI-PD [1], and Envelope Q-Learning [82]. The hyperparameters were those found to work well in the most similar discrete domains in the benchmark suite[1]. Due to computational constraints, the comparisons then focused on GPI-LS for scaling up to larger action spaces because (1) it has the best recorded results in this kind of domain [23], and (2) none of the other MORL methods in the suite were able to outperform GPI-LS in the experiments. Note that the more sophisticated GPI-PD yields essentially the same results as GPI-LS in this discrete context and action domain.

In short, even the baseline multi-objective evolution method strongly outperforms MORL (Fig. 7,8). The reason is that evolution inherently recombines blocks of knowledge, whereas MORL techniques struggle when there is no clear gradient of improvement.

## C  Pandemic Response Challenge

This section details the methods used in the application of RHEA to the XPRIZE Pandemic Response Challenge.

### C.1  Distillation

In distillation [3, 30, 31], the goal is to fit a model with a fixed functional form to capture the behavior of each initial solution, by solving the following minimization problem:

$$\theta_i^* = \underset{\theta_i}{\mathrm{argmin}} \int_{\mathcal{Q}} p(q) \left\| \pi_i(q) - \hat{\pi}_i\big(\kappa(q, \pi_i(q), \phi); \theta_i\big) \right\|_1 dq \tag{2}$$

$$\approx \underset{\theta_i}{\mathrm{argmin}} \frac{1}{n_q} \sum_{j=1}^{n_q} \left\| \pi_i(q_j) - \hat{\pi}_i\big(\kappa(q, \pi_i(q_j), \phi); \theta_i\big) \right\|_1, \tag{3}$$

---

[1] https://github.com/LucasAlegre/morl-baselines

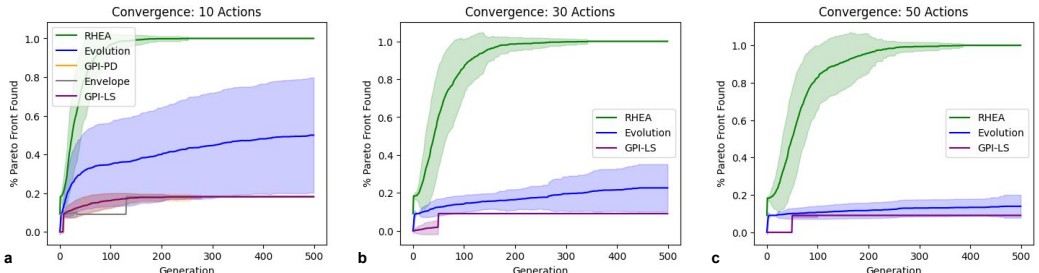

Figure 7: *Convergence curve comparisons.* **a-c,** Convergence curves for 10, 30, and 50 actions, respectively, in the Illustrative domain. RHEA converges to the full Pareto front in all cases, whereas the other methods converge to lower values as the action space grows. Evolution substantially outperformed the MORL baselines in all cases. With 10 actions, all MORL baselines converged relatively quickly to the same performance. Due to computational limitations, only the most relevant comparison, GPI-LS (which is state-of-the-art in discrete domains), was run in the experiments with more actions (lines are means; shading is standard deviation).

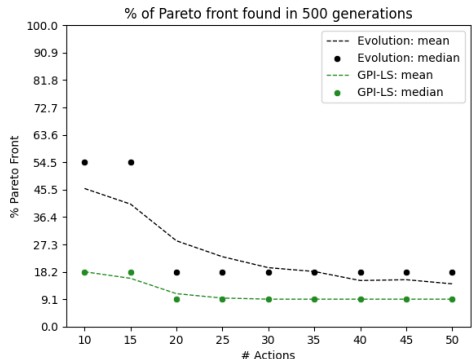

Figure 8: *MORL scaling comparison.* GPI-LS discovered less of the true Pareto front than the Evolution baseline (100 trials each). The performance of both methods decreases as the problem complexity, i.e., the number of actions, increases. This plot complements Fig. 6b. Recall from Fig. 6a that RHEA discovers the entire Pareto front in all trials.

where $q \in \mathcal{Q}$ is a query, $\pi_i$ is the initial solution, $\hat{\pi}_i$ is the distilled model with learnable parameters $\theta_i$, and $\kappa$ is a function that maps queries (which may be specified via a high-level API) to input data, i.e., *contexts*, with a canonical form that can be used to train $\hat{\pi}_i$. In practice, $\hat{\pi}_i$ is trained by optimizing $\theta_i$ with stochastic gradient descent using data derived from the $n_q$ queries for which data is available.

In the Pandemic Response Challenge experiment, prescriptors were distilled into an evolvable neural network architecture based on one previously used to evolve prescriptors from scratch in this domain [50], with the following changes: (1) In addition to the IPs used in that previous work, new IPs were used that were released in the Oxford data set since that work [27, 74], and that were used in the XPRIZE Pandemic Response Challenge; (2) Instead of a case growth rate, the case data input to the models were presented as cases per 100K residents. This input was found to allow distilled models to fit the training data more closely than the modified growth rate used in previous work. The reason for this improvement is that cases per 100K gives a more complete picture of the state of the pandemic; the epidemiological-model-inspired ratio used in prior work captures the rate of change in cases explicitly but makes it difficult to deduce how bad an outbreak is at any particular moment. Since many diverse submitted prescriptors took absolute case numbers into account, including these values in the distillation process allows the distilled prescriptors to align with their source model more closely.

Data for training a distilled model $\hat{\pi}_i$ was gathered by collecting the prescriptions made by $\pi_i$ in the XPRIZE Pandemic Response Challenge. Data was gathered for all prescriptions made with uniform

Afghanistan, Albania, Algeria, Andorra, Angola, Argentina, Aruba, Australia, Austria, Azerbaijan, Bahamas, Bahrain, Bangladesh, Barbados, Belarus, Belgium, Belize, Benin, Bermuda, Bhutan, Bolivia, Bosnia and Herzegovina, Botswana, Brazil, Brunei, Bulgaria, Burkina Faso, Burundi, Cambodia, Cameroon, Canada, Cape Verde, Central African Republic, Chad, Chile, China, Colombia, Comoros, Congo, Costa Rica, Cote d'Ivoire, Croatia, Cuba, Cyprus, Czech Republic, Democratic Republic of Congo, Denmark, Djibouti, Dominica, Dominican Republic, Ecuador, Egypt, El Salvador, Eritrea, Estonia, Eswatini, Ethiopia, Faeroe Islands, Fiji, Finland, France, Gabon, Gambia, Georgia, Germany, Ghana, Greece, Greenland, Guam, Guatemala, Guinea, Guyana, Haiti, Honduras, Hong Kong, Hungary, Iceland, India, Indonesia, Iran, Iraq, Ireland, Israel, Italy, Jamaica, Japan, Jordan, Kazakhstan, Kenya, Kosovo, Kuwait, Kyrgyz Republic, Laos, Latvia, Lebanon, Lesotho, Liberia, Libya, Lithuania, Luxembourg, Macao, Madagascar, Malawi, Malaysia, Mali, Mauritania, Mauritius, Mexico, Moldova, Monaco, Mongolia, Morocco, Mozambique, Myanmar, Namibia, Nepal, Netherlands, New Zealand, Nicaragua, Niger, Nigeria, Norway, Oman, Pakistan, Palestine, Panama, Papua New Guinea, Paraguay, Peru, Philippines, Poland, Portugal, Puerto Rico, Qatar, Romania, Russia, Rwanda, San Marino, Saudi Arabia, Senegal, Serbia, Seychelles, Sierra Leone, Singapore, Slovak Republic, Slovenia, Solomon Islands, Somalia, South Africa, South Korea, South Sudan, Spain, Sri Lanka, Sudan, Suriname, Sweden, Switzerland, Syria, Taiwan, Tajikistan, Tanzania, Thailand, Timor-Leste, Togo, Trinidad and Tobago, Tunisia, Turkey, Uganda, Ukraine, United Arab Emirates, United Kingdom / England, United Kingdom / Northern Ireland, United Kingdom / Scotland, United Kingdom / Wales, United Kingdom, United States / Alabama, United States / Alaska, United States / Arizona, United States / Arkansas, United States / California, United States / Colorado, United States / Connecticut, United States / Delaware, United States / Florida, United States / Georgia, United States / Hawaii, United States / Idaho, United States / Illinois, United States / Indiana, United States / Iowa, United States / Kansas, United States / Kentucky, United States / Louisiana, United States / Maine, United States / Maryland, United States / Massachusetts, United States / Michigan, United States / Minnesota, United States / Mississippi, United States / Missouri, United States / Montana, United States / Nebraska, United States / Nevada, United States / New Hampshire, United States / New Jersey, United States / New Mexico, United States / New York, United States / North Carolina, United States / North Dakota, United States / Ohio, United States / Oklahoma, United States / Oregon, United States / Pennsylvania, United States / Rhode Island, United States / South Carolina, United States / South Dakota, United States / Tennessee, United States / Texas, United States / Utah, United States / Vermont, United States / Virginia, United States / Washington, United States / Washington DC, United States / West Virginia, United States / Wisconsin, United States / Wyoming, United States, Uruguay, Uzbekistan, Vanuatu, Venezuela, Vietnam, Yemen, Zambia, Zimbabwe

Figure 9: List of the 235 geos (i.e., countries and subregions) whose data (from the Oxford dataset [27, 54, 74]) was used in XPRIZE competition and in experiments in this paper.

IP weights. This data consisted of ten date ranges, each of length 90 days, and 235 geos (Fig 9), resulting in 212,400 training samples for each prescriptor, a random 20% of which was used for validation for early stopping. More formally, each (date range, geo) pair defines a query $q$, with $\pi_i(q) \in \mathbb{Z}_5^{90 \times 12}$ the policy generated by $\pi_i$ for this geo and date range. The predicted daily new cases for this geo and date range given this policy is $\phi(q, \pi_i(q)) \in \mathbb{R}^{90}$. Let $\mathbf{h}$ be the vector of daily historical new cases for this geo up until the start of the date range. This query leads to 90 training samples for $\hat{\pi}_i$: For each day $t$, the target is the prescribed actions of the original prescriptor $\pi_i(q)_t$, and the input is the prior 21 days of cases (normalized by 100K residents) taken from $\mathbf{h}$ for prior days before the start of the date range and from $\phi(q, \pi_i(q))$ for days in the date range.

Distilled models were implemented and trained in Keras [7] using the Adam optimizer [35]. Mean absolute error (MAE) was used as the training loss (since policy actions were on an ordinal scale), with targets normalized to the range [0, 1]. The efficacy of distillation was confirmed by computing the rank correlations between the submitted expert models in the XPRIZE challenge and their distilled counterparts with respect to the two objectives: For both cases and cost, the Spearman correlation was $\approx 0.7$, with $p < 10^{-20}$, demonstrating that distillation was successful. In such a real-world scenario, a correlation much closer to 1.0 is unlikely, since many solutions are close together in objective space, and may have different positions on the Pareto front depending on the evaluation context.

## C.2 Evolution

In the Pandemic Response Challenge experiment, the evolution component was implemented inside using the Evolutionary Surrogate-assisted Prescription (ESP) framework [24], which was previously used to evolve prescriptors for IP optimization from scratch, i.e., without taking advantage of distilled models [50]. The distillation above results in evolvable neural networks $\hat{\pi}_1 \ldots \hat{\pi}_{n_\pi}$ which approximate $\pi_1 \ldots \pi_{n_\pi}$, respectively. These distilled models were then placed into the initial population of a run of ESP, whose goal is to optimize actions given contexts. In ESP, the initial population (i.e., before any evolution takes place) usually consists of neural networks with randomly generated weights. By replacing random neural networks with the distilled neural networks, ESP starts from diverse high-quality solutions, instead of low-quality random solutions. ESP can then be run as usual from this starting point.

In order to give all distilled models a chance to reproduce, the "population removal percentage" parameter was set to 0%. Also, since the experiments were run as a quantitative evaluation of teams in the XPRIZE competition [10, 11, 12], distilled models were selected for reproduction inversely

proportional to the number of submitted prescriptors for that team. This inverse proportional sampling creates fair sampling at the team level.

Baseline experiments were run using the exact same algorithm but with initial populations consisting entirely of randomly initialized models (i.e., instead of distilled models). The population size was 200; in RHEA, 169 of the 200 random NNs in the initial population were replaced with distilled models. Ten independent evolutionary runs of 100 generations each were run for both the RHEA and baseline settings.

The task for evolution was to prescribe IPs for 90 days starting on February 12, 2021, for the 20 regions with the most total deaths at that time. Internally, ESP uses the Pareto-based selection mechanism from NSGA-II to handle multiple objectives [16].

The current experiments were implemented with ESP because it is an already established method in this domain. Note, however, that such distillation followed by injecting in the initial population could be used in principle to initialize the population of any multi-objective evolution-based method that evolves functions.

### C.3 Pareto-based Performance Metrics

This section details the multi-objective performance evaluation approach used in this paper. It is based on comparing Pareto fronts, which are the standard way of quantifying progress in multi-objective optimization. While there are many ways to evaluate multi-objective optimization methods, the goal in this paper is to do it in a manner that would be most useful to a real-world decision-maker. That is, ideally, the metrics should be interpretable and have immediate implications for which method would be preferred in practice.

In the Pandemic Response Challenge experiment, each solution generated by each method $m$ in the set of considered methods $M$ yields a policy with a particular average daily cost $c \in [0, 34]$ and a corresponding number of predicted new cases $a \geq 0$ [50]. Each method returns a set of $N_m$ solutions which yield a set of objective pairs $S_m = \{(c_i, a_i)\}_{i=1}^{N_m}$. Following the standard definition, one solution $s_1 = (c_1, a_1)$ is said to *dominate* another $s_2 = (c_2, a_2)$ if and only if

$$(c_1 < c_2 \wedge a_1 \leq a_2) \vee (c_1 \leq c_2 \wedge a_1 < a_2), \tag{4}$$

i.e., it is at least as good on each metric and better on at least one. If $s_1$ dominates $s_2$, we write $s_1 \succeq s_2$. The Pareto front $F_m$ of method $m$ is the subset of all $s_i = (c_i, a_i) \in S_m$ that are not dominated by any $s_j = (c_j, a_j) \in S_m$. The following metrics are considered:

**Hypervolume Improvement (HVI)**  Dominated hypervolume is the most common general-purpose metric used for evaluating multi-objective optimization methods [61]. Given a reference point in the objective space, it is the amount of dominated area between the Pareto front and the reference point. The reference point is generally chosen to be a "worst-possible" solution, so the natural choice in this paper is the point with maximum IP cost and number of cases reached when all IPs are set to 0. Call this reference point $s_o = (c_o, a_o)$. Formally, the hypervolume is given by

$$\text{HV}(m) = \int_{\mathbb{R}^2} \mathbb{1}\Big[\exists\, s_* \in F_m : s_* \succeq s \wedge s \succeq s_o\Big] ds, \tag{5}$$

where $\mathbb{1}$ is the indicator function. Note that HV can be computed in time linear in the cardinality of $F_m$. HVI, then, is the improvement in hypervolume compared to the Pareto front $F_{m_o}$ of a reference method $m_o$:

$$\text{HVI(m)} = \text{HV}(m) - \text{HV}(m_o). \tag{6}$$

The motivation behind HVI is to normalize for the fact that the raw hypervolume metric is often inflated by empty unreachable solution space.

**Domination Rate (DR)**  This metric is a head-to-head variant of the "Domination Count" metric used in Phase 2 evaluation in the XPRIZE, and goes by other names such as "Two-set Coverage" [61]. It is the proportion of solutions in the Pareto front $F_{m_o}$ of reference method $m_o$ that are dominated by solutions in the Pareto front of method $m$:

$$\text{DR}(m) = \frac{1}{|F_{m_o}|} \cdot \Big|\{s_o \in F_{m_o} : (\exists\, s \in F_m : s \succeq s_o)\}\Big|. \tag{7}$$

The above generic multi-objective metrics can be difficult to interpret from a policy-implementation perspective, since, e.g., hypervolume is in units of cost times cases, and the domination rate can be heavily biased by where solutions on the reference Pareto front tend to cluster. The following three metrics are more interpretable and thus more directly usable by users of such a system.

**Maximum Case Reduction (MCR)**    This metric is the maximum reduction in number of cases that a solution on a Pareto front gives over the reference front:

$$\text{MCR}(m) = \max \left\{ a_o - a_* \; \forall \; (s_o = (c_o, a_o) \in F_{m_o}, \; s_* = (c_*, a_*) \in F_m) : s_* \succeq s_o \right\}. \quad (8)$$

In other words, there is a solution in $F_{m_o}$ such that one can reduce the number of cases by $\text{MCR}(m)$, with no increase in cost. If MCR is high, then there are solutions on the reference front that can be dramatically improved.

The final two metrics, RUN and REM, are instances of the R1 metric for multi-objective evaluation [28, 61], which is abstractly defined as the probability of selecting solutions from one set versus another given a distribution over decision-maker utility functions.

**R1 Metric: Uniform (RUN)**    This metric captures how often a decision-maker would prefer solutions from one particular Pareto front among many. Say a decision-maker has a particular cost they are willing to pay when selecting a policy. The RUN for a method $m$ is the proportion of costs whose nearest solution on the combined Pareto front $F_*$ (the Pareto front computed from the union of all $F_m \; \forall \; m \in M$) belong to $m$:

$$\text{RUN}(m) = 1/c_{\max} - c_{\min} \int_{c_{\min}}^{c_{\max}} \mathbb{1} \left[ \underset{s_* \in F_*}{\arg\min} \left\| c - c_* \right\| \in F_m \right] dc, \quad (9)$$

where $s_* = (c_*, a_*)$. Here, $c_{\min} = 0$, and $c_{\max} = 34$, since that is the sum of the maximum settings across all IPs. Note that RUN can be computed in time linear in the cardinality of $F_*$.

RUN gives a complete picture of the preferability of each method's Pareto front, but is agnostic as to the real preferences of decision-makers. In other words, it assumes a uniform distribution over cost preferences. The final metric adjusts for the empirical estimations of such preferences, so that the result is more indicative of real-world value.

**R1 Metric: Empirical (REM)**    This metric adjusts the RUN by the real-world distribution of cost preferences, estimated by their empirical probabilities $\hat{p}(c)$ at the same date across all geographies of interest:

$$\text{REM}(m) = \int_{c_{\min}}^{c_{\max}} \hat{p}(c) \cdot \mathbb{1} \left[ \underset{s_* \in F_*}{\arg\min} \left\| c - c_* \right\| \in F_m \right] dc. \quad (10)$$

In this paper, $\hat{p}(c)$ is estimated with Gaussian Kernel Density Estimation (KDE; Fig. 3d), using the scipy implementation with default parameters [75]. For the metrics that require a reference Pareto front against which performance is measured (HVI, DR, and MCR), Distilled is used as this reference; it represents the human-developed solutions, and the goal is to compare the performance of Human+AI (i.e. RHEA) to human alone.

All of the above metrics are used to compare solutions in Fig. 3 of the main paper. They all consistently demonstrate that RHEA creates the best solutions and that they also would be likely to be preferred by human decision makers.

### C.4   Analysis of Schedule Dynamics

The data for the analysis illustrated in Fig. 4 is from all submitted prescriptors and single runs of RHEA, evolution alone, and real schedules. Each point in Fig. 4a corresponds to a schedule $S \in \mathbb{Z}_5^{90 \times 12}$ produced by a policy for one of the 20 geos used in evolution. For visualization, each $S$ was reduced to $\hat{S} \in [0, 5]^{12}$ by taking the mean of IPs across time, and these 12-D $\hat{S}$ vectors were processed via UMAP [45] with `n_neighbors=25, min_dist=1.0`, and all other parameters default.

Below are the formal definitions of the high-level behavioral measures computed from $S$ and used in Fig. 4b. Let $S^+ \in [0, 34]^{90}$ be the total cost over IPs in $S$ for each day.

Swing measures the range in overall stringency of a schedule:

$$\text{Swing}(S) = \max_{i,j} \left| S_i^+ - S_j^+ \right|. \tag{11}$$

Separability measures to what extent the schedule can be separated into two contiguous phases of differing overall stringency:

$$\text{Separability}(S) = \max_t \frac{\left| \frac{1}{t} \sum_{i=0}^{t-1} S_i^+ - \frac{1}{90-t} \sum_{j=t}^{89} S_j^+ \right|}{\frac{1}{2} \left( \frac{1}{t} \sum_{i=0}^{t-1} S_i^+ + \frac{1}{90-t} \sum_{j=t}^{89} S_j^+ \right)}. \tag{12}$$

Focus increases as the schedule uses a smaller number of IPs:

$$\text{Focus}(S) = 12 - \sum_k \mathbb{1}(\hat{S}_k > 0). \tag{13}$$

Agility measures how often IPs change:

$$\text{Agility}(S) = \max_k \sum_{t=1}^{89} \mathbb{1}(S_{tk} \neq S_{(t-1)k}). \tag{14}$$

Periodicity measures how much of the agility can be explained by weekly periodicity in the schedule:

$$\text{Periodicity}(S) = \max \left( 0, \max_k \frac{\sum_{t=1}^{82} \mathbb{1}(S_{tk} \neq S_{(t-1)k}) - \sum_{t=7}^{89} \mathbb{1}(S_{tk} \neq S_{(t-7)k})}{\sum_{t=1}^{82} \mathbb{1}(S_{tk} \neq S_{(t-1)k})} \right). \tag{15}$$

These five measures serve to distinguish the behavior of schedules generated by different sets of policies at an aggregate level.

The violin plots in Fig. 4b were created with Seaborn [76], using default parameters aside from `cut=0`, `scale='width'`, and `linewidth=1` (https://seaborn.pydata.org/generated/seaborn.violinplot.html). The violin plots have small embedded boxplots for which the dot is the median, the box shows the interquartile range, and the whiskers show extrema.

### C.5   Pareto Contributions

To measure the contribution of individual models, the ancestry of individuals on the final Pareto front of RHEA is analyzed. For each distilled model $\hat{\pi}_i$, the number of final Pareto front individuals who have $\hat{\pi}_i$ as an ancestor is counted, and the percentage of genetic material on the final Pareto front that originally comes from $\hat{\pi}_i$ is calculated. Formally, these two metrics are computed recursively. Let $\text{Par}(\pi)$ be the parent set of $\pi$ in the evolutionary tree. Individuals in the initial population have an empty parent set; individuals in further generations have two parents. Let $F$ be the set of all individuals on the final Pareto front. Then, the ancestors of $\pi$ are

$$\text{Anc}(\pi) = \begin{cases} \varnothing & \text{Par}(\pi) = \varnothing, \\ \bigcup_{\pi' \in \text{Par}(\pi)} \text{Anc}(\pi') \cup \text{Par}(\pi) & \text{otherwise,} \end{cases} \tag{16}$$

and the Pareto contribution count is

$$\text{PCCount}(\pi) = |\{\pi' : \pi \in \text{Anc}(\pi') \text{ and } \pi \in F|, \tag{17}$$

while the percentage of ancestry of $\pi$ due to $\pi'$ is

$$\text{APercent}_{\pi'}(\pi) = \begin{cases} 0 & \text{Par}(\pi) = \varnothing, \pi \neq \pi', \\ 1 & \text{Par}(\pi) = \varnothing, \pi = \pi', \\ \frac{1}{|\text{Par}(\pi)|} \sum_{\pi'' \in \text{Par}(\pi)} \text{APercent}_{\pi'}(\pi'') & \text{otherwise,} \end{cases} \tag{18}$$

with the Pareto contribution percentage

$$\text{PCPercent}(\pi) = \frac{1}{|F|} \sum_{\pi' \in F} \text{APercent}_\pi(\pi'). \tag{19}$$

In the experiments, these two metrics are highly correlated, so only results for PCPercent are reported (Fig. 5).

## C.6 Energy Estimates

The relatively compact model size in RHEA makes it accessible and results in a low environmental impact. Each run of evolution in the Pandemic Response experiments ran on 16 CPU cores, consuming an estimated $3.9 \times 10^6$J. This computation is orders-of-magnitude less energy intensive than many other current AI systems: For instance, training AlphaGo took many limited-availability and expensive TPUs, consuming $\approx 8.8 \times 10^{11}$J [13]; training standard image and language models on GPUs can consume $\approx 6.7 \times 10^8$J [84] and $\approx 6.8 \times 10^{11}$J [77], respectively. Specifically:

- Each training run of RHEA for the Pandemic Response Challenge experiments takes $\approx$9 hours on a 16-core m5a.4xlarge EC2 instance. At 100% load, this instance runs at $\approx$120W (`https://engineering.teads.com/sustainability/carbon-footprint-estimator-for-aws-instances/`), yielding a total of $\approx 3.9 \times 10^6$J.

- The energy estimate of training AlphaGo was based on `https://www.yuzeh.com/data/agz-cost.html`, with 6380 TPUs running at 40W for 40 days, yielding a total of $\approx 8.8 \times 10^{11}$J.

- The energy estimate for image models is based on training a ResNet-50 on ImageNet for 200 epochs on a Tesla M40 GPU. The training time is based on https://arxiv.org/abs/1709.05011 [84]; the energy was computed from `https://mlco2.github.io/impact/`.

- The energy estimate for language models was based on an estimate of training GPT-3 (See Appendix D in https://arxiv.org/pdf/2007.03051.pdf [77]).

- Each training run of RHEA in the Illustrative Domain takes only a few minutes on a single CPU.

# D Ethics

This section considers ethical topics related to deploying RHEA and similar systems in the real world.

*Fairness.* Fairness constraints could be directly incorporated into the system. RHEA's multi-objective optimization can use any objective that can be computed based on the system's behavior, so a fairness objective could be used if impacts on the subgroups can be measured. A human user might also integrate this objective into their calculation of a unified Cost objective, since any deviation from ideal fairness is a societal cost. In deployment, an oversight committee could interrogate any developed metrics before they are used in the optimization process to ensure that they align with declared societal goals.

*Governance and Democratic Accountability.* This is a key topic of Project Resilience [55], whose goal is to generalize the framework of the Pandemic Response Challenge to SDG goals more broadly. We are currently involved in developing the structure of this platform. For any decision-making project there are four main roles: Decision-maker, Experts, Moderators, and the Public. The goal is to bring these roles together under a unified governance structure.

At a high-level, the process for any project would be: the Decision-maker defines the problem for which they need help; Experts build models for the problem and make them (or data to produce Distilled versions) public; Moderators supervise (transparently) what Experts contribute (data, predictors or prescriptors); The Public comments on the process, including making suggestions on what to do in particular contexts, and on ways to improve the models (e.g., adding new features, or modifying objectives); Experts incorporate this feedback to update their models; after sufficient discussion, the Decision-maker uses the platform to make decisions, looking at what the Public has suggested and what the models suggest, using the Pareto front to make sense of key trade-offs; The Decision-maker communicates about their final decision, i.e., what was considered, why they settled on this set of actions, etc. In this way, key elements of the decision-making process are transparent, and decision-makers can be held accountable for how they integrate this kind of AI system into their decisions. By enabling a public discussion alongside the modeling/optimization process, the system attempts to move AI-assisted decision-making toward participative democracy grounded in science. The closest example of an existing platform with a similar interface is https://www.metaculus.com/home/, but it is for predictions, not prescriptions, and problems are not linked to particular decision-makers.

It is important that the public has access to the models via an "app", giving them a way to directly investigate how the models are behaving, as in existing Project Resilience proof-of-concepts[234], along with a way of flagging any issues/concerns/insights they come across. A unified governance platform like the one outlined above also would enable mechanisms of expert vetting by the public, decision-makers, or other experts.

The technical framework introduced in this paper provides a mechanism for incorporating democratically sourced knowledge into a decision-making process. However, guaranteeing that sourced knowledge is democratic is a much larger (and more challenging) civil problem. The concepts of power imbalances and information asymmetry are fundamental to this challenge. Our hope is that, by starting to formalize and decompose decision-making processes more clearly, it will become easier to identify which components of the process should be prioritized for interrogation and modification, toward the goal of a system with true democratic accountability.

For example, the formal decomposition of RHEA into Define, Gather, Distill, Evolve, enables each step to be interrogated independently for further development. The implementation in the paper starts with the most natural implementation of each step as a proof-of-concept, which should serve as a foundation for future developments. For example, there is a major opportunity to investigate the dynamics of refinements of the Distill step. In the experiments in this paper, classical aggregated machine learning metrics were used to evaluate the quality of distillation, but in a more democratic platform, experts could specify exactly the kinds of behavior they require the distillation of their models to capture. By opening up the evaluation of distillation beyond standard metrics, we could gain a new view into the kinds of model behavior users really care about. That said, methods could also be taken directly from machine learning, such as those discussed App. A. However, we do not believe any of these existing methods are at a point where the humans can be removed from the loop in the kinds of real-world domains the approach aims to address.

*Data Privacy and Security.* Since experts submit complete prescriptors, no sensitive data they may have used to build their prescriptors needs to be shared. In the Gather step, each expert team had an independent node to submit their prescriptors. The data for the team was generated by running their prescriptors on their node. The format of the data was then automatically verified, to ensure that it complied with the Defined API. Verified data from all teams was then aggregated for the Distill & Evolve steps. Since the aggregated data must fit an API that does not allow for extra data to be disclosed, the chance of disclosing sensitive data in the Gather phase is minimized. One mechanism for improving security is to allow the user of each role to rate sources, data, and models for a quality, reliability, and security standpoint, similar to established approaches in cybersecurity[5].

*External Oversight.* Although the above mechanisms all could yield meaningful steps in addressing a broad range of ethical concerns, they cannot completely solve all issues of ethical deployment. So, it is critical that the system is not deployed in an isolated way, but integrated into existing democratic decision-making processes, with appropriate external oversight. Any plan for deployment should include a disclosure of these risks to weigh against the potential societal benefits.

*Sustainability and Accessibility.* See App. C.6 for details on how energy usage estimates were computed.

# E   Data Availability

The data collected from the XPRIZE Pandemic Response Challenge (in the Define and Gather phases) and used to distill models that were then Evolved can be found on AWS S3 at `https://s3.us-west-2.amazonaws.com/covid-xprize-anon` (i.e., in the public S3 bucket named 'covid-xprize-anon', so it is also accessible via the AWS command line). This is the raw data from the Challenge, but with the names of the teams anonymized. The format of the data is based on the format developed for the Oxford COVID-19 Government Response Tracker [27].

---

[2]`https://evolution.ml/demos/npidashboard/`

[3]`https://climatechange.evolution.ml/`

[4]`https://landuse.evolution.ml/`

[5]https://www.first.org/global/sigs/cti/curriculum/source-evaluation

# F   Code Availability

The formal problem definition, requirements, API, and code utilities are for the XPRIZE, including the standardized predictor, are publicly available [10, 9]. The prediction and prescription API, as well as the standardized predictor used in the XPRIZE and the evolution experiments can be found at `https://github.com/cognizant-ai-labs/covid-xprize`. The Evolve step in the experiments was implemented in a proprietary implementation of the ESP framework, but the algorithms used therein have been described in detail in prior work [24]. Code for the illustrative domain was implemented outside of the proprietary framework and can be found at `https://github.com/cognizant-ai-labs/rhea-demo`.

