# OpenReview forum: "Unlocking the Potential of Global Human Expertise"
_NeurIPS.cc/2024/Conference — NeurIPS 2024 poster_

### Official Review · Reviewer_CmcD · 2024-07-06

**Soundness:** 3
**Presentation:** 3
**Contribution:** 2
**Rating:** 5
**Confidence:** 2

**Summary:**

This paper designs a framework called Realizing Human Expertise through AI (RHEA) to combine and distill solutions from a diverse set of models or solutions provided by experts. The steps are: Define Problem -> Gather Solutions -> Distill Model -> Evolve Solution. Examples were presented to demonstrate the framework's effectiveness in different domains.

**Strengths:**

The author proposes a potential solution to collective knowledge that can be combined to discover solutions. It could be a way to find a better solution from the information stream when there is more than one solution.

**Weaknesses:**

1. Some explanations of the framework are unclear and require additional details to better explain the "Distill" process.
2. The contribution paragraph needs more information to clearly describe the real contribution.
3. The conclusion is too broad and needs to be more specific.

**Questions:**

1) The concern is the efficiency of this framework, as the Gather step requires waiting for experts to respond. The framework involves a human-in-the-loop process. How can bias be reduced when gathering solutions from the experts? (if these experts' solutions are biased or limited)
2) During a crisis, what happens if there are insufficient solutions? If there are two opposing solutions, what will be the outcome?
3) The description of the Distill process lacks sufficient detail for the reader to understand how it works.

**Limitations:**

This work has limited real-world evaluation. Fairness also needs to be considered in developing this framework.

---

> ### Author Rebuttal · Authors · 2024-08-07
>
> _Response to question on details of the Distill process:_
>
> We will move the core details of Distill from the Appendix to the main text and ensure that there is enough in the main text to have a clear idea of the process.
>
> _Response to suggestions to add more specific details to the Contributions and Conclusion paragraphs:_
>
> Thank you for the suggestion. We will revise both these paragraphs to focus more on the technical contributions of the paper, specifically, details of the framework, implementation, and experiments.
>
> _Response to questions around the effects of having humans playing a role in the process:_
>
> These are very interesting practical considerations. This work focuses on the technical aspect of how to set up the framework, but the problem of how to improve efficiency in gathering expert solutions is a social/civil problem. It could be improved e.g. through improved communication/solicitation methods. Such considerations are outside the scope of this paper, but constitute interesting avenues of future work, as will be noted in the discussion.
>
> On the other hand, RHEA has a natural method to overcome bias in expert solutions, since, through the process of evolution, the objectively highest quality components can recombine and persist. If expert knowledge is limited, RHEA will simply revert to the evolution baseline.
>
> _Response to: “During a crisis, what happens if there are insufficient solutions? If there are two opposing solutions, what will be the outcome?”_
>
> This is exactly the problem that RHEA is designed to solve! RHEA yields a broad Pareto front of solutions, to maximize the chance that one will be useful in any particular scenario.
>
> _Response to “This work has limited real-world evaluation”:_
>
> While further real-world evaluation would always be useful, this paper actually includes an unusually substantial such evaluation already. The COVID-19 intervention application required soliciting, incentivizing, and gathering the input of over 100 teams of experts in over 20 countries. Such a real-world demonstration of the Define and Gather steps at scale is a major and unique contribution of this work. Further, the optimization results were evaluated with an expansive dataset of real-world pandemic interventions and outcomes in over 200 countries and two years—the first such dataset to be created. We will clarify these contributions in the paper.
>
> _Response to “Fairness also needs to be considered in developing this framework.”:_
>
> RHEA has mechanisms for fairness, by assigning probabilities to expert knowledge for recombination, and providing traces of where contributed knowledge comes from. However, we agree that it is possible for fairness issues to enter the system, especially if actions with bad intentions submit solutions (as also outlined in the response to Reviewer 1). Detecting and preventing such adversarial behavior is an interesting avenue for future work.

---

> > ### Comment · Reviewer_CmcD · 2024-08-10
> >
> > Thank you to the authors for responding to my questions. Their explanations should be considered for inclusion in the paper.

---

> > > ### Author Response · Authors · 2024-08-12
> > >
> > > Thank you for appreciating our explanations. Would you consider increasing your score based on the impact of these explanations, along with the new comparisons to state-of-the-art MORL methods and theoretical justification?

---

### Official Review · Reviewer_fdTu · 2024-07-16

**Soundness:** 3
**Presentation:** 3
**Contribution:** 3
**Rating:** 6
**Confidence:** 2

**Summary:**

The authors introduce a new evolutionary framework for combining expert policy predictions called RHEA and evaluate performance in both a synthetic domain that is highly interpretable and in a predictor from the XPRIZE Pandemic Response Challenge. The solutions are analyzed by qualitatively and quantitatively and compared against less sophisticated baselines.

**Strengths:**

The paper is clearly written and focuses on an important problem. I liked that the authors explain the framework in the context of a highly interpretable example and then do significant qualitative and quantitative analysis on the solutions discovered by in the XPRIZE domain.

**Weaknesses:**

See questions.

**Questions:**

I did not get the sense that the “policy intervention problem” is something that is well studied. How is the problem statement here different from a contextual bandit or similar formalisms (knapsack problems?)
Are there other baselines for this problem besides lesioned versions of the current framework?
What are the computational differences between RHEA and the baselines was each algorithm run until convergence? Is it possible to show learning curves?

**Limitations:**

Yes

---

> ### Author Rebuttal · Authors · 2024-08-07
>
> > I did not get the sense that the “policy intervention problem” is something that is well studied. How is the problem statement here different from a contextual bandit or similar formalisms (knapsack problems?) Are there other baselines for this problem besides lesioned versions of the current framework? What are the computational differences between RHEA and the baselines was each algorithm run until convergence? Is it possible to show learning curves?
>
> *Response to Questions above:*
>
> These clarifications are indeed useful; thank you for pointing them out. The problem that the Evolve portion of the framework solves is formulated as a general decision-making problem. It can indeed be framed for multi-objective reinforcement learning (MORL) by casting the predictor as an environment that prescriptors interact with as they learn. We ran such an experiment for this response, and describe the results in our Main Response. In short, the state-of-the-art MORL methods do not do well on these kinds of problems because they have a hard time recombining blocks of knowledge in useful ways, whereas this process is natural in evolutionary methods.
>
> In the pdf attachment to our Main Response, we provide learning curves for RHEA, Evolution, and the MORL methods. Each was given the same number of evaluations, i.e., calls to the predictor. In terms of wall-clock time, RHEA and Evolution were also much faster than the MORL methods, especially model-based methods that require expensive gradient updates. The theoretical analysis in the Main Response also addresses questions of method convergence.

---

> > ### Comment · Reviewer_fdTu · 2024-08-08
> >
> > Thank you for the response. I have no further questions.

---

> > > ### Author Response · Authors · 2024-08-09
> > >
> > > Great! We are glad we were able to address all you questions. Would you be willing to increase your score based on the information provided, especially the new comparisons to state-of-the-art MORL methods and the new theoretical justification? Thanks!

---

> > > > ### Comment · Reviewer_fdTu · 2024-08-09
> > > >
> > > > I'm positively inclined towards the paper, but I'm not confident enough in my assessment to be the champion for its acceptance.

---

### Official Review · Reviewer_R6Y1 · 2024-07-27

**Soundness:** 3
**Presentation:** 3
**Contribution:** 2
**Rating:** 5
**Confidence:** 3

**Summary:**

This paper introduces RHEA (Realizing Human Expertise through AI), a framework for combining diverse human expertise to solve complex problems using artificial intelligence. The key contributions are:
* Recognizing the challenge of integrating diverse human expertise to solve global problems like those in public health.
* Identifying requirements for an AI process to effectively combine human expertise.
* Proposing the RHEA framework to meet these requirements, consisting of four steps: 1) Define the problem formally 2) Gather diverse expert solutions 3) Distill solutions into a canonical form (ie neural networks) 4) Evolve the distilled solutions to discover improved solutions
* Demonstrating RHEA on a synthetic example.
* Applying RHEA to a real-world problem: optimizing COVID-19 intervention policies.
* Analyzing, for that real world problem: how RHEA recombines and innovates upon human expertise to discover improved solutions.

The paper shows (at least for their examples) that RHEA can discover broader and more effective policy strategies than either AI or human experts alone. It highlights RHEA's ability to realize latent potential in diverse human expertise, even from solutions that may not seem immediately useful. The authors argue this approach could help bridge the gap between human-only and AI-only decision-making for complex global problems.

**Strengths:**

Originality: The paper presents a novel framework, RHEA (Realizing Human Expertise through AI), which creatively combines human expertise with an evolutionary framework. This approach is original in several ways: The framework introduces a process of distilling human-created solutions into a canonical form (neural networks) that can then be evolved using population-based search.

Quality: The authors provide a step-by-step explanation of the RHEA framework, though it’s unclear to me to how immediately useful those steps are as they are quite high level.
* The framework is tested first on a synthetic example making it easier to understand.
* The analysis of results on the COVID example is thorough, using multiple performance metrics and visualizations to support their claims.
* The authors acknowledge some of the limitations and potential issues

Clarity: The paper is generally well-structured and clearly presented:
* The paper is well written.
* The introduction effectively sets up the problem and the paper's contributions.
* The illustration is helpful for explaining the concept.
* The use of figures, particularly in explaining the RHEA framework and visualizing results, aids in understanding.

Significance: The paper's significance lies in its potential impact on AI research and applications:
* It addresses the important challenge of leveraging human expertise in AI systems, which is crucial for tackling complex global problems.
* The RHEA framework could be applied to a wide range of domains beyond COVID-19 policy optimization.
* The work demonstrates a practical approach to combining diverse expert knowledge, which could be valuable for collaborative problem-solving in various fields.

**Weaknesses:**

Here's a substantive assessment of the weaknesses of the paper:

* Limited generalizability: The paper's primary application focuses on COVID policy optimization, which, while relevant, may not sufficiently demonstrate the framework's broad applicability. The authors could strengthen their claims by: a) Providing a theoretical analysis for why RHEA should work in other domains. b) Discussing potential applications in other fields, with specific examples of how RHEA might be adapted.

* Novelty: The evolutionary algorithm comes from other work so the idea is basically to initialize the parameters with distilled neural networks. This is a powerful way to solve this specific problem but given the limited analysis of the algorithm both empirically (when does it work) and theoretically, I am not sure if this work should be published at an ML conference or if it would be better at a more general purpose venue.

* Insufficient comparison to existing methods: While the paper compares RHEA to evolution alone and distilled models, it lacks comparison to other state-of-the-art methods in policy optimization or ensemble learning. The authors should: a) Include comparisons to relevant baseline methods (e.g., reinforcement learning approaches, other multi-objective optimization techniques). b) Discuss how RHEA compares to other methods of combining expert knowledge, such as boosting or stacking.

* Generalizability concerns: The paper does not sufficiently address what the operating regime is of RHEA. What problems does it work? What's the scale at which it works? Does it work if you have thousands of experts?

* Lack of Theoretical guarantees: While the empirical results are promising, the paper lacks theoretical analysis of RHEA's properties. The authors could strengthen the paper by: a) Providing theoretical bounds on the performance of RHEA under certain conditions. b) Analyzing the convergence properties of the evolutionary process. c) Discussing how the distillation process affects the solution space and optimization landscape.

* The structure of the paper: Some important details are in the appendix while  there is a lot of analysis of the specific solutions and details (like the carbon footprint)  which should be in the appendix. For example, as the specific ways this work is technically novel are unclear, the authors should move some of the related work to the main text.

Addressing these weaknesses would significantly strengthen the paper, providing a more comprehensive and robust presentation of the RHEA framework and its potential impact.

Post rebuttal update:

**Questions:**

* How confident are you that RHEA can be applied effectively to domains beyond COVID-19 policy optimization? Could you provide examples of other complex problems where you believe RHEA would be particularly effective, and explain why? And also where it would not be effective?

* Have you conducted any comparisons between RHEA and other state-of-the-art methods in policy optimization or ensemble learning? If so, what were the results? If not, which methods do you think would be most relevant for comparison?

* How does the computational complexity of RHEA scale with the number of experts and the complexity of the problem space? Are there any foreseeable bottlenecks in applying RHEA to much larger or more complex problems?

* Have you conducted any theoretical analysis of RHEA's properties, such as convergence guarantees or performance bounds? If so, could you elaborate on these? If not, what kind of theoretical results do you think would be most valuable to pursue?

* How interpretable are the solutions produced by RHEA compared to the original expert inputs? Can you provide a detailed case study of a specific evolved solution, explaining how it combines and improves upon the original expert knowledge?

* Could you provide more details on the distillation process? How do you ensure that the neural network accurately captures the essence of the expert's solution, especially for complex or nuanced strategies? For example, you find a Spearman correlation of 0.7 which does not seem particularly high to me. Also, I assume this 0.7 on the test set? What is the spearman correlation on the training set?

* Does RHEA have mechanisms to incorporate new expert knowledge or feedback over time? How adaptable is the framework to changing conditions or new information?

**Limitations:**

The authors discuss the potential for approximation errors when applying the policies to the real world.
The paper also mentions the need for user studies to fully evaluate the real-world effectiveness of RHEA prescriptors. The authors recognize that their cost measure was uniform over interventions, which may not reflect real-world scenarios.

I think this is generally a good set of limitations. Potential areas where the authors could improve:
* The authors could explore more deeply the potential long-term societal impacts of widespread adoption of AI-assisted policy-making tools like RHEA.
* While briefly mentioned, the paper could elaborate on how to ensure the RHEA process remains transparent and interpretable to non-expert stakeholders.
* The limitations section could address how RHEA might handle rapidly changing conditions or unexpected scenarios not covered in the initial expert knowledge.

---

> ### Author Rebuttal · Authors · 2024-08-07
>
> Our responses to remaining comments are grouped by topic:
>
> _NeurIPS vs. general science venue:_
>
> We agree that this approach could be well-appreciated by broad audiences. Indeed, NeurIPS encourages application papers that have broad appeal, and we believe the introduction of this framework to the NeurIPS community could catalyze ML researchers to consider practical concerns that lead to interesting ML problems to solve. Consequently, in the revision we will add additional comparisons to multi-objective reinforcement learning (MORL) methods, expand the theoretical analysis, and showcase technical details and context in the main paper, thus strengthening this impact.
>
> _Comparisons:_
>
> W.r.t. comparisons to ensembling methods, the MoE comparison in the Illustrative Domain (Figure 2) serves as a multi-objective Oracle for stacking, since it shows the performance of a stacker that optimally selects experts based on context. Boosting does not apply to our problem: it relies on experts having access to each other’s models, whereas we assume that the expert models are submitted in a single traunch. We will clarify these points in the paper. This also relates to your question on adaptively adding new experts during optimization, which is discussed below.
>
> _Paper structure:_
>
> Thank you for this feedback. We will move the most relevant portions of the related work to the main text, specifically the discussion of multi-objective reinforcement learning and ensembling methods. We will also move technical methodological details to the main paper, such as those on distillation, to make the main paper technically self-contained.
>
> _Scaling to more complex problems:_
>
> The population-based search used by RHEA extends readily to thousands of experts; the theoretically optimal population size is often quite large [4, 5]. We will note these details in the final version of the paper.
>
> Bottlenecks could come from adversarial attacks in submissions based on overwhelming the system with non-useful experts. In this paper we have assumed that the humans submitting solutions are well-intentioned. We will mention this potential concern in the discussion.
>
> _Interpretable case study:_
>
> First, the illustrative domain provides a clear case study exactly along the lines suggested. Second, in the COVID-19 domain we get at this quantitatively through the topological analysis of the final solutions (Figure 4) and evolutionary dynamics (Figure 5). Figure 4 illustrates several complete schedules generated from RHEA solutions, and Figure 5 shows how evolved solutions in RHEA solutions tend to lie between their parents in objective space. We will add a detailed case study of a single solution from Figure 4 and its expert parents to make the interpretation more detailed and concrete.
>
> _Distillation process:_
>
> The process is detailed in the Appendix; details will be moved to the main paper. While other implementations are possible, this one is simple, intuitive, and was shown to work well based on the quantitative results in the overall framework. In the future, we can investigate different distillation processes and their effects on the overall framework. In this kind of real-world scenario, correlation much closer to 1.0 is unlikely, since many solutions are close together in objective space, and may have different positions on the Pareto front depending on the evaluation context.
>
> Full autoregressive schedule roll-outs were not generated in the Distill step, as the models were trained on next-step prediction. The mean MAE over distilled models on both the training and validation splits of the distillation dataset was ~0.1, i.e., the mean difference between the actual and predicted actions (which could range from 0-5) was small.
>
> _Online adaptation:_
>
> This is a great avenue for future work! The present implementation of RHEA naturally supports the incorporation of new expert knowledge over time, since it maintains a population of solutions over time, so new expert solutions could at any time be distilled and inserted into the populations. Since it is likely that the objective quality of the new knowledge falls somewhat behind the current Pareto front, a mechanism might be required to ensure the new knowledge survives for long enough in the population to be effectively exploited.
>
> It would also be possible to update the predictor based on new data, as has been demonstrated in earlier work on evolutionary surrogate-assisted prescription [Francon et al 2020], and even increase the number of context or action variables online (e.g., in the case of pandemic interventions, new kinds of interventions could be implemented).
>
> _Long term societal impacts:_
>
> This is a very interesting social question. One risk with AI-assisted policy-making tools is that as human users come to rely on them more there may be less incentive to gather useful and diverse knowledge from human experts. The hope with RHEA is that, by centering the value of soliciting and gathering human expert knowledge, humans will be rewarded for the perspectives they share, and it will encourage expert diversity to thrive. We will clarify this discussion of risk/reward trade-off in the discussion of the paper.
>
> _Transparency:_
>
> RHEA can already assign contribution percentages of submitted experts to final solutions, as depicted in Figure 5. One can further imagine an interface, where upon selecting a solution for inspection, a decision-maker is alerted to who contributed to this solution, at which point they have the opportunity to investigate the contribution, e.g., raise a flag if many selected solutions have the majority of their contribution from a single actor or small group of actors. Analysis of such a scenario could reveal outstanding objective expertise by this group, or some unintended bias in the system. We will clarify this point in the discussion.
>
>
> Thank you again for articulating these issues. We agree that our response strengthens the work and its impact.

---

### Author Rebuttal · Authors · 2024-08-07

Main Response:

Thank you to the reviewers for the feedback. We were glad the reviewers agreed that the work was valuable, and suggested how it could be strengthened further by addressing some outstanding questions. This main response focuses on three main points brought up by multiple reviewers:

1. Comparisons: We have performed further experimental comparisons to multi-objective reinforcement learning (MORL) methods, highlighting how they fall short, creating a research opportunity for RHEA.
2. Theory: We have provided theoretical justification for the main methodology of evolutionary optimization.
3. Generalizability: We have clarified the scope of the method and detailed how it can be applied to other domains.

**Comparisons**

In response to questions about additional baselines, we performed comparisons to a suite of MORL techniques [1] in the Illustrative domain. We ran preliminary tests with several of the recent algorithms, namely, GPI-LS [2], GPI-PD [2], and Envelope Q-Learning [3]. Due to computational constraints, we then focused on GPI-LS for scaling up to larger action spaces because (1) it has the best recorded results in this kind of domain [1], and (2) none of the other MORL methods in the suite were able to outperform GPI-LS in our experiments.

In short, the baseline multi-objective evolution method strongly outperforms MORL (see plots in rebuttal pdf). The reason is that evolution inherently recombines blocks of knowledge, whereas MORL techniques struggle when there is no clear gradient of improvement. We will add this discussion and the code for the MORL comparisons to the paper and its online supplement. Further, we will move the most critical portions of Related Work from the Appendix to the main paper to clarify the motivation for these comparisons.

**Theory**

The theory will depend on the particular implementation of the RHEA framework. For this paper, we can provide a theoretical analysis based on recent convergence analysis of NSGA-II [4, 5], which is the multi-objective evolutionary algorithm used in the paper.
In theoretical settings, the performance of this algorithm has been shown to depend critically on the size of “jumps” in the optimization landscape, roughly, the maximum size of non-convex regions in the landscape. When these regions are minimal, the method converges to the full ground truth Pareto front in $O(N n \lg n)$ evaluations. When the jump size increases, the best known bound is $O(N^2 n^k / \Theta(k)^k)$, where $k$ is a measure of the jump size, $n$ is the problem dimensionality, and $N$ is the population size. In other words, an increasing jump size causes a roughly exponential slowdown of convergence.

Distilling useful, diverse experts can be viewed as a way of decreasing the jump size. This process is clearly shown in the illustrative domain, where the experts provide building blocks that can be immediately recombined to discover better solutions, but that are difficult to discover from scratch. This interpretation is borne out in the experiments, as RHEA continues to converge quickly as the action space (i.e. problem dimensionality) increases, whereas evolution regresses to only being able to discover the most convex (easily-discoverable) portions of the Pareto front.

We will articulate this theoretical framework and the requisite assumptions fully in the revision.

**Generalizability**

RHEA can be applied effectively to policy-discovery domains where (1) the problem can be formalized with contexts, actions, and outcomes; (2) there exist diverse experts from which solutions can be gathered; and (3) the problem is sufficiently challenging. In contrast, RHEA would not be effective, (1) if the problem is too easy, so that the input from human experts would not be necessary; (2) if the problem is hard, but no useful and diverse experts exist; (3) if there is no clear way to define context and/or action variables upon which the experts agree. These aspects of generality will be clarified in the revision.

The modularity of the overall framework means that different implementations of components can be designed for different domains, such as those related to sustainability, engineering design, and public health. One particularly exciting opportunity for RHEA mentioned several times in the paper is climate policy. The paper gives the example of defining “green hydrogen” as a concrete example of the kind of challenge RHEA could solve, but this could be just a small part of a climate policy application. For example, the En-Roads climate simulator supports diverse _actions_ across energy policy, technology, and investment; _contexts_ based on social, economic, and environmental trajectories; and multiple competing _outcomes_, including global temperature, cost of energy, and sea-level rise (https://en-roads.climateinteractive.org/). Users craft policies based on their unique priorities and expertise. RHEA could be used with a predictor like En-Roads to discover optimized combinations of expert climate policies that trade-off across temperature change and other the outcomes that users care about most. We will add this discussion to the paper.



A point-by-point response to remaining comments in each review is provided below. The same reference numbers are used throughout the responses.

[1] Felten, et al. “A Toolkit for Reliable Benchmarking and Research in Multi-Objective Reinforcement Learning”. NeurIPS 2023. https://github.com/LucasAlegre/morl-baselines.

[2] Alegre, et al. “Sample-Efficient Multi-Objective Learning via Generalized Policy Improvement Prioritization”. AAMAS 2023.

[3] Yang, Sun, and Narasimhan. “A Generalized Algorithm for Multi-Objective
Reinforcement Learning and Policy Adaptation”. NeurIPS 2019.

[4] Doerr and Qu. “From Understanding the Population Dynamics of the NSGA-II to the First Proven Lower Bounds”. AAAI 2023.

[5] Doerr and Qu. “Runtime Analysis for the NSGA-II: Provable Speed-Ups from Crossover”. AAAI 2023.

---

### Decision · Program_Chairs · 2024-09-25

**Decision:**

Accept (poster)

**Comment:**

This paper proposes a framework for aggregating expertise to help inform policymaking with steps: Define Problem -> Gather Solutions -> Distill Model -> Evolve Solution.

All reviewers agreed that the approach was interesting, novel, and potentially impactful, and appreciated the exposition around the real-world importance of the problem. There were significant questions about whether the approach would generalize beyond the specific question where it was tested, but reviewers were willing to overlook this since the area as a whole is quite new and has such clear  connections to so many other areas of emerging importance for NeurIPS like democratic accountability, social welfare measurement, etc.